# Significance of Phosphate Nano-Fertilizers Foliar Application: A Brief Real-Field Study of Quantitative, Physiological Parameters, and Agro-Ecological Diversity in Sunflower

**Dávid Ernst** [1], **Marek Kolenčík** [1,*], **Martin Šebesta** [2], **Ľuba Ďurišová** [3], **Samuel Kšiňan** [3], **Lenka Tomovičová** [3], **Nikola Kotlárová** [3], **Mária Kalúzová** [3], **Ivan Černý** [1], **Gabriela Kratošová** [4], **Veronika Žitniak Čurná** [1], **Jana Ivanič Porhajašová** [3], **Mária Babošová** [3], **Edmund Dobročka** [5], **Yu Qian** [6], **Sasikumar Swamiappan** [7], **Ramakanth Illa** [8], **Shankara Gayathri Radhakrishnan** [9], **B. Ratna Sunil** [10] **and Ladislav Ducsay** [1]

1 Institute of Agronomic Sciences, Faculty of Agrobiology and Food Resources, Slovak University of Agriculture in Nitra, Tr. A. Hlinku 2, 94976 Nitra, Slovakia; david.ernst@uniag.sk (D.E.)
2 Institute of Laboratory Research on Geomaterials, Faculty of Natural Sciences, Comenius University in Bratislava, Mlynská Dolina, Ilkovičova 6, 84215 Bratislava, Slovakia
3 Institute of Plant and Environmental Sciences, Faculty of Agrobiology and Food Resources, Slovak University of Agriculture in Nitra, Tr. A. Hlinku 2, 94976 Nitra, Slovakia
4 Nanotechnology Centre, CEET, VŠB-Technical University of Ostrava, 17. listopadu 15/2172, 70800 Ostrava, Czech Republic
5 Institute of Electrical Engineering, Slovak Academy of Sciences, Dúbravská cesta 9, 84104 Bratislava, Slovakia
6 School of Ecology and Environmental Science, Yunnan University, 2 Cuihubei Lu, Kunming 650091, China
7 Department of Chemistry, VIT University, Vellore 632014, India
8 Department of Chemistry, School of Advanced Sciences, VIT-AP University, Amaravati 522237, India
9 Department of Chemistry, University of Pretoria, Pretoria 0002, South Africa
10 Department of Mechanical Engineering, Bapatla Engineering College, Bapatla 522101, India
* Correspondence: marek.kolencik@uniag.sk

**Abstract:** One of the challenges in agriculture practices is guaranteeing an adequate and bioavailable phosphorus supply for plants on phosphorus-deficient soils. A promising alternative lies in the utilization of phosphate nano-fertilizers (NFs) through spray applications. Therefore, this short-term study aimed to investigate the yet undetermined widespread impact of P-NFs on crops characterized by broad leaves, an intensive rate of photosynthesis, and belonging to the oilseed plant, sunflower (*Helianthus annuus* L.). To achieve this, NFs were applied at lower concentrations of various phosphate-based NFs, including (i) nano-hydroxylapatite (nano-Hap) and (ii) a mixture of nano-calcium zinc phosphate and macro-sized parascholzite (nano/macro-ZnPhos), in comparison to the NF-free control. The study was carried out under authentic field conditions during the 2022 vegetation season at the Dolná Malanta site within the Central European Region. The empirical evidence presented herein indicates that the utilization of biocompatible and bioactive nano-Hap, initially engineered for biomedical applications, and nano/macro-ZnPhos, now foliarly applied at reduced concentrations, elicited a statistically significant elevation in quantitative parameters and seasonal physiological responses. The parameters analyzed included head diameter, dry head weight, seed yield per hectare, nutritional seed oiliness, etc. as well as the physiological normalized difference vegetation index (NDVI), stomatal conductance index (Ig), and crop water stress index (CWSI). In terms of agro-ecological terrestrial bio/diversity, it was evident that the nano/macro-ZnPhos was the most hospitable variant for the terrestric insect community, but surprisingly, the agronomically more popular nano-Hap showed only statistically insignificant changes in the diversity of the detected communities. However, the relevance of outcomes highlighted using nano-fertilizers, supporting the concept of precision and sustainable agriculture under field conditions.

**Keywords:** agronomy; phosphate nano-fertilizers; plant nutrients; hydroxylapatite; sunflower; foliar application; field conditions; quantitative parameters; physiological parameters; agro-ecological diversity

## 1. Introduction

The progress in nanotechnology is beginning to integrate fields such as agriculture, where the development of nano-fertilizers (NFs) is one of the most significant advancements. The primary objective in NF development is the reduction in traditional fertilizers to nanoscale dimensions [1]. Applied NFs follow a definition similar to that of nanomaterials (NMs), where one of the three dimensions should not exceed 100 nm; otherwise, it becomes a mixture of NF with micro-sized fertilizers [2–4]. Significantly, the size of nanoparticles (NPs) in NFs, compared to their analogical chemical macro-sized or ionic counterparts, implicates different reactivity, mobility, and bioavailability [5].

In the context of agricultural experimentation, the sunflower, belonging to the Asteraceae family, emerges as an ideal candidate for foliar NF application—a domain that remains underexplored in academic studies. The sunflower, globally renowned as a premier oil-bearing crop [6], boasts broad leaves with a surface morphology well-suited for the deposition of NFs [7,8].

Sunflowers demonstrate an exceptional capacity to absorb and distribute metal-related NPs or their residues through their leaves [9,10], holding promise for applications in bioremediation strategies [11].

Liu and Lal [1] introduced a definition of NFs and classified them according to their essentiality and the functions for plants. These categories encompass: (i) macronutrient-based NFs containing elements such as Ca, N, P, and K; (ii) micronutrient-based NFs featuring elements such as Cu, Fe, Zn, etc.; NMs augmented with nutrients, typically associated with porous silica or layered silicates; and (iv) plant-growth nano-stimulators without obvious essentiality to plant that exhibited yet not fully described mechanisms of action, such as carbon nanotubes, $TiO_2$ NPs, etc. Despite this comprehensive classification, there remains a gap in the literature concerning the combined effects of NFs from different categories on agronomical outcomes. While recent works by Sharma et al. [12] and Sharma et al. [13] have indicated research involving nano-phosphorus combined with other nutrients, the broader interaction between diverse NF categories and their agronomical impacts remains relatively unexplored.

Phosphorus (P) stands as one of the essential macronutrients vital for supporting normal plant growth and development. Its application in agronomy, particularly to crops such as sunflower, aims to facilitate the optimal development of the root system, particularly concerning nutrient and water uptake from the soil environment [14].

Additionally, P plays a pivotal role in regulating plant energy generation through ATP synthesis and various transport mechanisms [15,16], enhances resistance to environmental stresses, and intensifies physiological reactions [17] while improving overall growth and development [13] as well as increasing crop productivity [18,19]. However, the extensive application of P in agricultural settings can disrupt its natural geochemical cycle, potentially giving rise to imbalanced microorganism populations and concerns of eutrophication [20]. An unresolved question pertains to its impact on the complex biodiversity of terrestrial agro-ecological communities within agricultural environments.

NFs offer a potential solution by allowing the use of lower concentration ranges and facilitating more precise targeting, thus leading to higher nutrient use efficiency and more direct agronomic benefits. A notable example is apatite, the most common naturally occurring compound containing P with various crystal modifications such as hydroxylapatite (Hap), chlorapatite and fluorapatite. During dissolution processes, apatite can release either P or calcium cation, $Ca^{2+}$ [21]. Ca, recognized as an essential macronutrient, plays a pivotal role in various physiological processes. Within an optimal concentration range, Ca supports proper signal transduction for photosynthesis [22], contributes to cell wall formation and fortification [23], regulates enzymatic and metabolic activities [24], and enhances resistance against stress factors [25].

While it is widely acknowledged that Ca (and B) are critical for the progressive development of sunflowers [26], Zn is also needed for optimal plant growth [27] and quality of production [28]. The application of P through spray deposition offers several distinct

advantages, primarily the direct leaf contact that enables precise concentration application, leading to rapid plant responses. Oppositely, in comparison to a soil environment, P could be inappropriately immobilized or imperfectly released from phosphorus-soil minerals [29,30]. Thus, foliar application provides better access to P for plants in P-deficient soil, where bioavailable P has to be supplied with annual-periodic frequency. Nowadays, a certain inquiry pertains to the sustainable acquisition of ample supply of P while mitigating significant environmental impact and loss of agronomical functionality. Regarding phosphate compounds, methods including apatite formation though precipitation [31], sol–gel synthesis [32], or electrodeposition [33] were published. A biotechnological approach for their production has also been reported [34,35].

In this context, the utilization of secondary food waste derived from animal sources to produce Hap carries significant advantages. This approach not only reduces waste and its subsequent impact on landfills but also offers a more economically viable means of recycling compared to conventional industrial mining. However, a relatively minor drawback might arise from the chemical variability inherent in precursor biomaterials, such as fish bones [36], potentially affecting the quality of the final production and the precise effects of NFs.

The application of NFs through the soil, leaves, or stems is an important aspect to consider as the point of entry into plants relates to subsequent effects. In this regard, several relevant factors should be taken into consideration. The size of NFs influences their primary interaction with and subsequent entry into the plant. Moreover, the chemical composition and crystallinity of these NPs affect their initial stability and reactivity [37,38]. Morphology and surface energy also play a crucial role in, dictating the reactivity and solubility of the NFs. Furthermore, reactions occurring within the applied dispersion can involve pH changes and conductivity variations, impacting dissolution and nutrient bioavailability [39].

When considering the application of phosphorus-containing compounds, their environmental stability is primarily influenced by factors such as the chemical stoichiometry, the structure, the presence of ambient forces, and the ratio between cations and anions [40] as well as the crystallinity or the type of active solvents [41]. In the realm of agricultural practices, specifically concerning spray applications, there is a notable absence of information pertaining to the dissolution of P when combined with additional Zn ions or various chemical phases, including nano-sized particles. As a consequence, there is a need for the investigation of less frequently utilized phosphates such as parascholzite (Pslz) [42] or calcium zinc phosphate variants characterized by slightly differing chemical formulas, crystalline structures, and solubilities to facilitate their practical implementation in agronomy.

In the academic sphere, the current landscape tends to lean more toward laboratory research rather than providing a comprehensive understanding of the tangible effects of NFs on crops. Therefore, the aim of this short-term study is to elucidate the impact of foliar spray deposition of two types of phosphorus-based NFs on quantitative agronomically important indices, the physiology of sunflowers (*Helianthus annuus* L.), and terrestrial agro-ecological populations under field conditions during the 2022 vegetation season in the central European region.

## 2. Materials and Methods

### 2.1. Origin and Characterization of Sprayed Phosphorus Nano-Fertilizers Applied on Sunflower

2.1.1. Formation of Hydroxylapatite

To facilitate the formation of an innovative generation of phosphorus-based NFs, specifically hydroxylapatite, bone samples were sourced from Sheelavati fish (*Roho labio*) obtained from the Krishna River. Initially, the bones were boiled in water to remove any residual organic matter, which was followed by drying via hot air. The bones were then mechanically crushed into small fragments and placed in a glass container. Thermal

modification was carried out at 600 °C according to the methodology described by Sunil and Jagannatham [43].

### 2.1.2. Formation of Mixture Nano-Sized Calcium Zinc Phosphate and Macro-Sized Parascholzite

To produce a mixture of macro-sized parascholzite and nano-sized calcium zinc phosphate as NFs, a co-precipitation method was used. Initially, 0.425 M of calcium nitrate $(Ca(NO_3)_2 \cdot 4H_2O)$ and 0.075 M of zinc nitrate $(Zn(NO_3)_6 \cdot 6H_2O)$ were dissolved in 20 mL of deionized water. Subsequently, a 0.3 M phosphate solution $(NH_4)_2HPO_4$ was dissolved in 10 mL of deionized water. The pH solution was adjusted to the pH = 10 with 28% $NH_4OH$. Next, $(NH_4)_2HPO_4$ was reacted with a mixture of $Ca(NO_3)_2 4H_2O/Zn(NO_3)_6 \cdot 6H_2O$ to reach the steady state at room temperature. The subsequent mixture was continuously stirred at 800 rpm, 80 °C for 3 h. After that, the resulting final product—precipitate—was dried in a hot oven at 200 °C for 2 h and then subjected to heat at 1000 °C for 1 h. The electrical conductivity of the applied nano-fertilizer samples as well as the control variant was measured using the conductivity EC meter DiST5 (HANNA, RI, USA), and the pH values were analyzed using a pH meter inoLab 730 (Weilheim, Germany). The values are presented in Table S1 (Supplementary Materials).

### 2.2. Physico-Chemical Determination of Phosphorus Nano-Fertilizers

The JEOL 7610 F+ scanning electron microscopy (SEM) was used for visualization of the NPs' dimension and morphology (JEOL, Akishima, Tokyo, Japan), and the chemical composition of phosphorus-based NFs was examined by energy-dispersive X-ray spectrometry (EDAX) on a Phillips XL30 (EDS).

X-ray diffraction (XRD) analysis for crystallographic verification of the synthesized NFs and an assessment of structural parameters was conducted using the Bruker D8 DISCOVER diffractometer for X-ray diffraction (Bruker, Billerica, MA, USA). Measurements were taken applying a Cu anode at 300 mA, 40 kV, and 12 kW. The unit cell parameters were subsequently calculated using TOPAS 3.0 software (Bruker, MA, USA).

### 2.3. Plant Material

To conduct our field experiment, we used the SY Neostar hybrid (*Helianthus annuus* L.), a sunflower hybrid from Syngenta in Basel, Switzerland. This hybrid is specifically developed for use in the ClearField Plus® system of production, as it is resistant to two-line imidazoline. The SY Neostar variety is known for its medium to short height, exhibiting medium to early growth and development. It demonstrates adaptability to various environmental and growth conditions without specific agricultural requirements. Additionally, this hybrid offers an average oil content of approximately 47% and high resistance against *Sclerotinia sclerotiorum* and *Diaporte helianthi* while also displaying tolerance to *Plasmopara halstedii* [44].

### 2.4. Climate Seasonal Variation

According to meteoblue-data [45], the variability of precipitation (in mm) and average daily temperature (in °C) was evaluated during the 2022 vegetation season (Figure 1 and Table S2, Supplementary Materials).

The experimental study was conducted at the fields of Slovak University of Agriculture in Nitra (SUA Nitra) in Dolná Malanta, which is located close to Nitra in the Slovak Republic, Central European Region (48°19′25.41′′ N 18°09′2.89′′ E). Locality is elevated 250 m above sea level, and experimental fields are located in the northeastern part of Podunajská Lowland, close to Žitavská Hillock, south of the Zobor Hill (Tribeč mountain range). From a petrological point of view, the locality's bedrocks consist of granite and Mesosoic carbonates as well as Neogene and Quarternal development with eluvial sediments [46]. The soil in this area has been classified as silt loam haplic Luvisol [47] with the soil minerals content corresponding to quartz, muscovite, and anorthite [28]. The experimental field

belongs to maize cultivation with an intensive soil farming practice, and at the time of the study, sunflowers were being grown as part of a 7-plot crop rotation system.

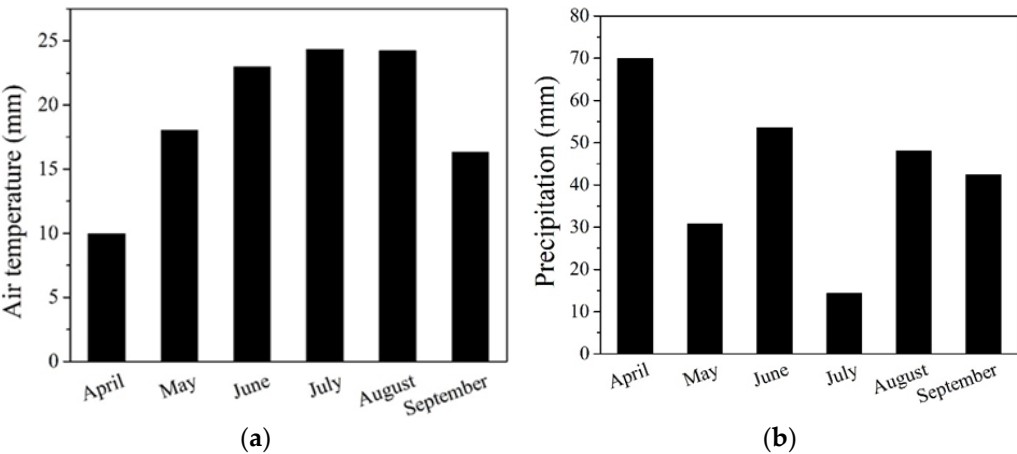

(**a**) (**b**)

**Figure 1.** Monthly variations from April to September in (**a**) air temperature and (**b**) precipitation during the 2022 vegetation season at the experimental field Dolná Malanta in Nitra, Slovakia, Central European Region.

### 2.5. Field Experiment Setup

The one-year field experiment was carried out using a method of three variants including a control randomly arranged in blocks selected perpendicularly, following the methodology described by Kolenčík et al. [8]. Each variant was replicated three times [48], and replication covered an area of 60 m². To prepare the field for the experiment, deep ploughing was carried out in autumn 2021 using a Zetor Forterra (Zetor Tractors, a.s. Brno, Czech Republic). Winter wheat (*Triticum aestivum* L.) was cultivated as the forecrop. Prior to sunflower sowing in spring 2022, the soil characteristics were analyzed, and the results are presented in Table 1.

**Table 1.** (**a**) Results of agrochemical soil sampling in autumn 2021. (**b**) Results of agrochemical soil sampling in spring 2022.

| | (a) | | | |
|---|---|---|---|---|
| **Year** | **P** | **K** | **Ca** | **Mg** |
| | **(mg kg$^{-1}$)** | | | |
| 2021 | 85 | 340 | 2500 | 304 |
| | (b) | | | |
| **Year** | **N$_{an}$** | **N-NO$_3^-$** | **N-NH$_4^+$** | **Dry matter** |
| | **(mg kg$^{-1}$)** | | | **%** |
| 2022 | 14.2 | 8.8 | 5.4 | 86.80 |

The 15-15-15 DUSLOFERT fertilizer (Duslo, a. s., Šaľa, Slovak Republic) was dosed at 400 kg.ha$^{-1}$ through machinery, specifically by the tractor, during pre-sowing soil cultivation, employing the Ferti fertilizer applicator (Agromehanica, Boljerac, Serbia).

In accordance with Albeiro et al. [49], sunflower seeds were sown in rows at a depth of 60 mm with a seed distance of 220 mm and an inter-row spacing of 700 mm, implementing the Monosem NG Plus 3 planter (Monosem, Largeasse, France). Alongside all treatments, including the NF-free control, the Wing® herbicide (BASF, Ludwigshafen am Rhein, Germany) was applied at a concentration of 4 dm³·ha$^{-1}$, and Pictor® fungicide was applied at a dose of 0.4 dm³ ha$^{-1}$ (BASF, Ludwigshafen am Rhein, Germany) by an AGT 865T/S sprayer (Agromehanica, Boljerac, Serbia).

Dispersions with phosphorus-based NFs were applied at a concentration of 33 mg·dm$^{-3}$. These dispersions were deposited onto the sunflower plants using a hand-held pressure sprayer (Mythos Di Martino, Mussolente, Italy). Three series of experiments were used:

(1) Without nano-fertilizers (only water);
(2) Nano-hydroxyapatite (nano-Hap);
(3) Nano-calcium zinc phosphate + macro-sized parascholzite (nano/macro-ZnPhos).

The application was carried out in the early morning under calm wind conditions until the sunflower leaves were completely wet. The first spray application took place on day 40 after sowing during the leaf development stage, while the second spray occurred on day 80 after sowing during the stem elongation stage and flower bud formation (Figure 2). Prior to each foliar deposition to sunflower, the NF samples were subjected to a 15-min ultrasonic treatment for better colloidal–dispersion properties [8].

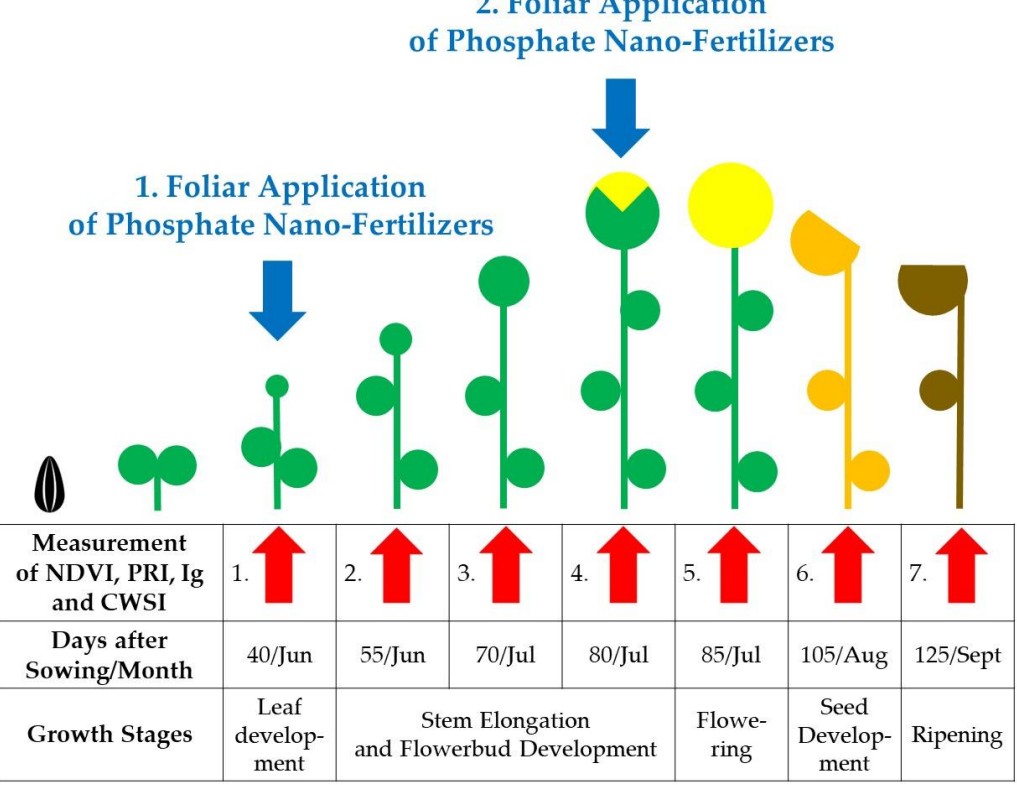

**Figure 2.** Schematic model of sunflower growth stages and spray-dispersion application period of phosphorus NFs and control without NF.

*2.6. Microscopic Analysis of Stomata of Sunflower Leaves after Phosphorus Nano-Fertilizers Treatments*

Sunflower leaves were collected for the purpose of investigating their structure—the stomata, following the second spray application during the flower-bud development. To facilitate the examination of the stomata, samples were prepared using the microrelief (replica) method. This involved applying diluted colorless nail polish with acetone onto the sunflower leaves, which subsequently created replicas of the stomata features when the duct tape was placed on a glass slide, adhering to Pazourek's technique [50]. Observation of the stomata included the adaxial and the abaxial side of the leaf using an Olympus BX41 light microscope with images captured by an Olympus E 520 camera (Shinjuku, Tokyo, Japan). Each treatment, including the control variant, was subjected to three repetitions. Within each repetition, ten random photographs of the leaf epidermis were taken. The evaluation encompassed the following parameters: stomata length and width, stomatal pore dimensions, and stomata count per 100,000 μm$^2$. The obtained data were subjected

to statistical analysis using the Tukey HSD test with a significance level set at $\alpha = 0.01$. Statistical computations were performed applying with TIBCO Statistica®, Version 14.0 (TIBCO Software Inc., Palo Alto, CA, USA) [51].

### 2.7. Evaluation of Quantitative Parameters of Sunflower

After the sunflowers reached full seed maturity, they were collected using a small pot and Claas Dominator 38 (CLAAS KGaA mbH, Harsewinkel, Germany). To determine the moisture level of the seeds, an analysis was carried out with the He Lite device (Pfeuffer GmbH, Kitzingen, Germany). Several quantitative parameters of sunflower were examined for each treatment, involving a manual count of plants with heads (pcs), measurement of head diameter (mm) with the Texi 4007 laboratory equipment (Texi GmbH, Berlin, Germany), measurement of head weight (g) and thousand seed weight (g) (TSW) using the Kern PCB3500-2 lab scale (KERN & Sohn GmbH, Balingen, Germany), and Numirex seed count (MEZOS spol. s.r.o., Hradec Králové, Czech Republic). The sunflower yield was calculated conveniently in tons per hectare (t ha$^{-1}$).

### 2.8. Analysis of Seasonal Physiology and Progress Physiological Reactions during Vegetation

To evaluate the normalized difference vegetation index (NDVI), we utilized the Plant-Pens NDVI 300 device (Photon System Instruments, Brno, Czech Republic). This device operates on the principle of analyzing the absorption of a reflected wavelength from the leaf surface. Specifically, we measured at the wavelengths of 740 nm and 660 nm for NDVI, which was subsequently calculated using the following Equation (1):

$$NDVI = (R_{740} - R_{660})/(R_{740} + R_{660}). \tag{1}$$

In order to ensure consistency and accuracy, all measurements were conducted on the same day—the equivalent plants, fully developed leaves, and with the analogical stage of development—between 11 a.m. and 1 p.m. Additionally, for precise determination of leaf heterogeneity, perpendicular measurement with a minimum of 10 points for each leaf per index were analyzed, as recommended by Gamon et al. [52].

For the analysis of stomatal conductance index (Ig) and crop water stress index (CWSI), the EasIR-4 thermo-camera was employed (Bibus AG, Fehraltorf, Switzerland). This analysis was conducted following the method described by Jones et al. [53], which integrated various leaf temperature parameters ($T_{leaf}$), involving dry ($T_{dry}$) and wet ($T_{wet}$) leaf surface temperatures, as well as atmospheric moisture. The Ig and CWSI were subsequently calculated with Equations (2) and (3):

$$Ig = (T_{dry} - T_{leaf})/(T_{leaf} - T_{wet}), \tag{2}$$

$$CWSI = (T_{leaf} - T_{wet})/(T_{dry} - T_{wet}). \tag{3}$$

Temperature images were captured by the thermo-camera diagonally on sunflower leaves from a distance of 2 m, at a height of 1.5 m, and with a resolution of $20.6° \times 15.5°$ in auto-focus mode.

### 2.9. Agro-Ecological Assessment

To collect insects with predominant epigeic components, the earth traps method was employed. This method involved the use of 1 dm$^3$ jars that were inserted directly into the soil and covered to protect them from rainfall and rodents in all treatments. At monthly intervals, the jars were controlled, and if necessary, refilled with a mixture consisting of 5% formaldehyde and 1/3 fixative solution according to Ernst et al. [28].

In order to determine the biological characteristics of collected insect specimens, special focus was given to the *Carabidae* family, which has a major role in biodiversity. For this group, elaboration and classification were completed systematically following the methods described in Pokorný's work [54].

To estimate the agro-ecological biodiversity, the abundance of the epigeic population and determination of *Coleoptera*, as well as obtained families within the *Carabidae* family, were taken into consideration. These groups also play important roles in assessing the overall terrestrial agro-ecological diversity.

### 2.10. Statistical Analysis

The statistical analysis for nonfertilizer treatment variants was applied with TIBCO Statistica®, version 14.0 (TIBCO Software Inc., Palo Alto, CA, USA) [51]. Prior to the evaluation of the multifactorial analysis of variance (ANOVA), the normality of experimental data was tested at $\alpha = 0.05$ and $\alpha = 0.01$ significance by Student's t-test, the Shapiro–Wilk test for trials, and Fisher's least significant difference (LSD).

## 3. Results

### 3.1. Physico-Chemical Characteristics of Prepared New Generation of Phosphate Nano-Fertilizers

In Figure 3a, the typical morphology of hydroxylapatite NPs is visible, predominantly spherical in shape, occasionally elongated in one direction corresponding to a pseudo-hexagonal shape, and mostly around 50 nm or lower in size, and individual particles are integrated into larger agglomerates. According to the EDS determination, the content of Ca, P and O correspond to the composition of pure hydroxylapatite (Figure 3b).

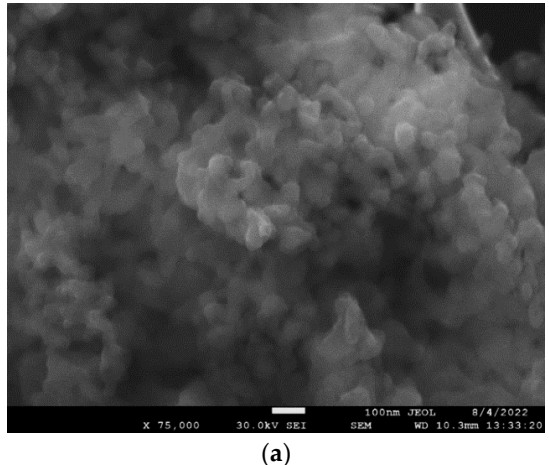 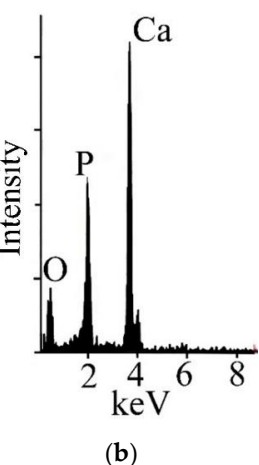

(**a**)　　　　　　　　　　　　　　　　　　　　　　　　(**b**)

**Figure 3.** (**a**) Typical scanning electron image (SEM) of hydroxylapatite NPs obtained through thermal treatment of Sheelavati fish bones and applied as a new generation of NFs; (**b**) Chemical verification of hydroxylapatite containing mainly Ca, P, and O elements visualized using electron-dispersive X-ray spectroscopy (EDS).

According to X-ray diffraction analysis (Figure S1, Supplementary Materials) and calculated detailed structural parameters (Table 2), we confirmed the production of hydoxylapatite.

**Table 2.** X-ray diffraction analysis of hydroxylapatite with calculated unit cell parameters.

| Chemical Formula | $Ca_{10}(PO_4)_6(OH)_2$ |
|---|---|
| Crystal symmetry | Hexagonal |
| *a*-axes | 9.46 |
| *b*-axes | 9.46 |
| *c*-axes | 6.88 |
| $\alpha$ | 90° |
| $\beta$ | 90° |
| $\gamma$ | 120° |
| Space group | $P6_3/m$ |
| Unit cell dimension | 615.7022 Å$^3$ (Calculated from Unit Cell) |
| Lvol-IB | $36.4 \pm 0.5$ nm (Calculated from X-ray diffraction data) |

In the case of the second applied NFs obtained by the chemical co-precipitation method, two morphological and size-distributed individuals were founded. The first type is predominantly spherical in shape with nano-size distribution of about 50 nm, and the second one corresponds to macro-particles incorporated into larger aggregates with leaf-shaped or prolongated table-forms of approximately 10–15 µm (Figure 4a) with a verified chemical composition of C, O, Zn, P, and Ca (Figure 4b).

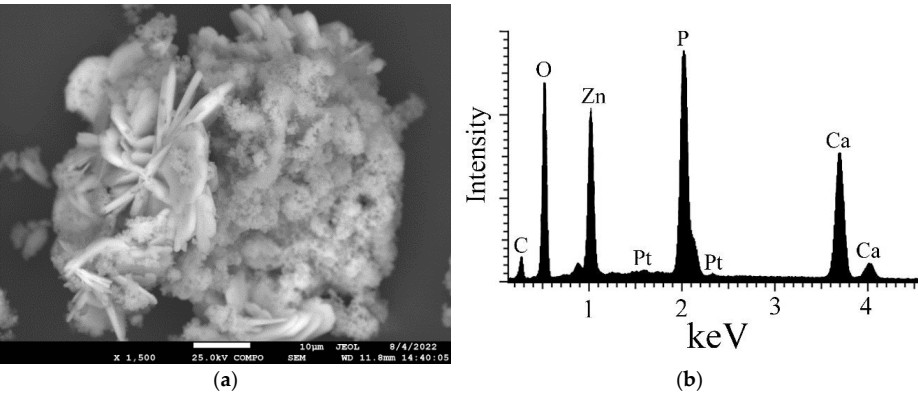

(a)　　　　(b)

**Figure 4.** (**a**) Typical scanning electron imaging (SEM) visualizing a mixture of macro-sized parascholzite and nano-sized calcium zinc phosphate formed by chemical co-precipitation route used as NFs; (**b**) EDS analysis from Figure 5a of new generation of phosphate-based NFs where the main elements such as C, O, Zn, P, and Ca were detected, which correspond to a parascholzite and calcium zinc phosphate mixture.

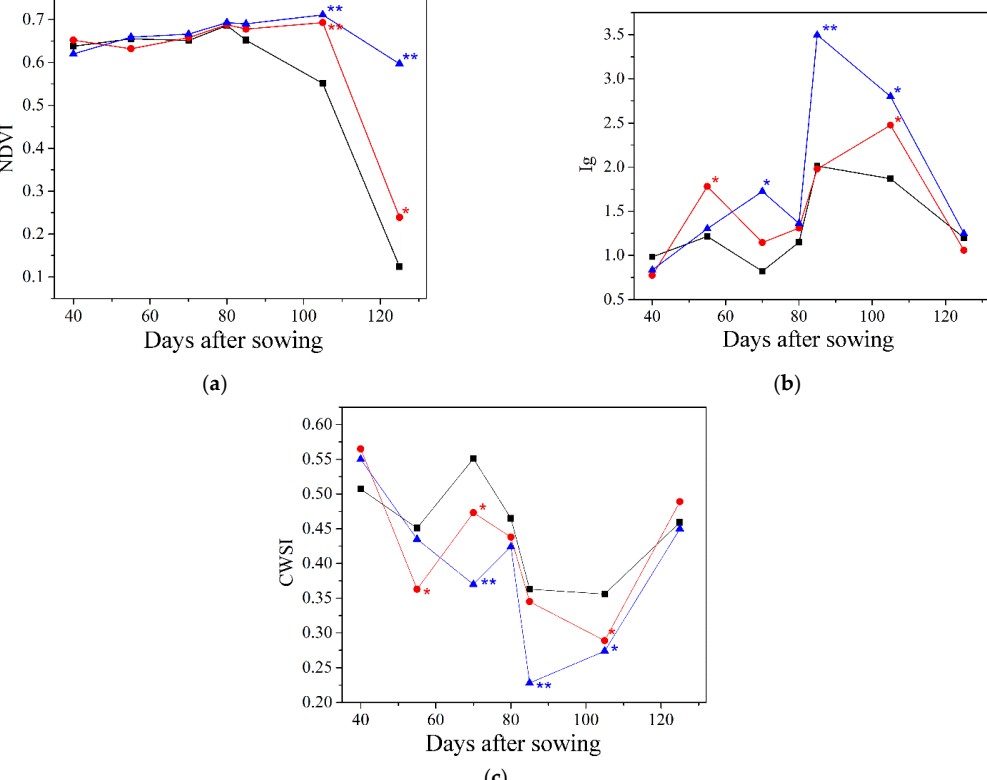

**Figure 5.** Evaluation of the progress of physiology under spray-deposited nano/macro-ZnPhos (shown as blue triangle), nano-Hap (red circle) treatment in contrast with NF-free control (black square) in the 2022 vegetation season based on physiological indices. (**a**) Normalized difference vegetation index (NDVI); (**b**) Stomatal conductance index (Ig); (**c**) Crop water stress index (CWSI); significance: * $p$ value < 0.05, ** $p$ value < 0.01.

Based on X-ray diffraction analysis (Figure S2), the formation of a mixture of macro-sized parascholzite and nano-dimensional calcium zinc phosphate is also confirmed along with precise structural parameters (Table 3).

**Table 3.** X-ray diffraction analysis of the mixture of parascholzite and calcium zinc phosphate with calculated unit cell parameters.

| | Parascholzite | Calcium Zinc Phosphate |
|---|---|---|
| Chemical formula | $CaZn_2(PO_4)_2 \cdot 2H_2O$ | $Ca_{19}Zn_2(PO_4)_{14}$ |
| Crystal symmetry | monoclinic | rhombohedral |
| *a*-axes | 17.864 | 10.3567 |
| *b*-axes | 7.422 | 10.3567 |
| *c*-axes | 6.674 | 37.173 |
| $\alpha$ | 90° | 90° |
| $\beta$ | 106.45 | 90° |
| $\gamma$ | 90° | 90° |
| Space group | C 2/c | R3c |
| Unit cell volume * | 884.883 Å$^3$ (Calculated from Unit Cell) | 3987.221 Å$^3$ (Calculated from Unit Cell) |
| Lvol-IB * | 130 ± 10 nm (Calculated from X-ray diffraction data) | 37 ± 3 nm (Calculated from X-ray diffraction data) |
| Content of minerals (%) ** | 57% | 43% |

* Calculated from Unit Cell. ** Calculated from X-ray diffraction analysis.

### 3.2. Analysis of Leave Surface Structures—Stomata after Foliar Treatment with Phosphorous Nano-Fertilizers

From the one-year results, it is evident that the number of stomata on both the adaxial and abaxial side of the sunflower leaf exhibited the same trend. The control variant (NF-free) had the highest number of stomata, which was followed by a slightly lower but statistically insignificant number of stomata in the nano-Hap application. The statistically significant lowest number of stomata was observed in the nano/macro-ZnPhos variant (Table 4).

**Table 4.** (**a**). Reaction of leaf structures—stomata and their parameters under phosphorus nano-fertilizers treatments on the upper side of the leaf—mean length and width of stomata, length and width of stomatal pores, number of stomata (N) per 100,000 μm$^2$ along with standard deviation (SD), coefficient of variation (CV), minimal and maximal values (Min, Max), superscript letters indicate the results of Tukey's HSD test ($\alpha$ = 0.01). (**b**). Reaction of leaf structures—stomata and their parameters under phosphorus NFs treatments on the lower side of the leaf—mean length and width of stomata, length and width of stomatal pores, number of stomata (N) per 100,000 μm$^2$ along with standard deviation (SD), coefficient of variation (CV), minimal and maximal values (Min, Max), superscript letters indicate the results of Tukey's HSD test ($\alpha$ = 0.01).

| (a) | | | |
|---|---|---|---|
| | Control Variant (No Application) | Nano-Hap (Foliar Applied Variant) | Nano/macro-ZnPhos (Foliar Applied Variant) |
| Length of stomata | | | |
| Mean ± SD (μm) | 26.40 ± 3.39 [a] | 26.61 ± 2.87 [a,b] | 27.46 ± 2.87 [b] |
| CV (%) | 12.84 | 10.80 | 10.46 |
| Min | 18.07 | 19.57 | 20.92 |
| Max | 33.70 | 34.44 | 35.47 |
| Width of stomata | | | |
| Mean ± SD (μm) | 15.00 ± 1.40 [a] | 15.09 ± 1.86 [a] | 15.03 ± 1.44 [a] |
| CV (%) | 9.30 | 12.30 | 9.56 |
| Min | 12.16 | 11.78 | 11.39 |
| Max | 18.78 | 19.42 | 18.84 |

**Table 4.** *Cont.*

|  | (a) | | |
|---|---|---|---|
|  | **Control Variant (No Application)** | **Nano-Hap (Foliar Applied Variant)** | **Nano/macro-ZnPhos (Foliar Applied Variant)** |
| **Length of stomatal pores** | | | |
| Mean ± SD (μm) | 16.53 ± 3.13 [a] | 16.71 ± 2.48 [a] | 16.23 ± 2.93 [a] |
| CV (%) | 18.91 | 14.86 | 18.04 |
| Min | 9.99 | 10.56 | 7.85 |
| Max | 24.49 | 23.68 | 22.68 |
| **Width of stomatal pores** | | | |
| Mean ± SD (μm) | 5.74 ± 1.40 [b] | 6.19 ± 1.02 [a] | 6.35 ± 1.05 [a] |
| CV (%) | 24.42 | 16.44 | 16.45 |
| Min | 2.72 | 3.15 | 4.00 |
| Max | 9.82 | 8.97 | 9.38 |
| **Number of stomata per 100,000 μm$^2$** | | | |
| Mean ± SD (μm) | 29.27 ± 4.98 [b] | 27.80 ± 3.70 [a,b] | 24.10 ± 5.57 [a] |
| CV (%) | 17.02 | 13.31 | 23.10 |
| Min | 22 | 20 | 14 |
| Max | 43 | 38 | 36 |
|  | (b) | | |
|  | **Control Variant (No Application)** | **Nano-Hap (Foliar Applied Variant)** | **Nano/macro-ZnPhos (Foliar Applied Variant)** |
| **Length of stomata** | | | |
| Mean ± SD (μm) | 28.58 ± 4.18 [a,b] | 29.42 ± 3.48 [b] | 27.79 ± 4.32 [a] |
| CV (%) | 14.63 | 11.82 | 15.54 |
| Min | 18.26 | 20.96 | 16.68 |
| Max | 38.67 | 38.52 | 38.92 |
| **Width of stomata** | | | |
| Mean ± SD (μm) | 17.03 ± 3.28 [a] | 18.61 ± 3.84 [b] | 15.96 ± 2.91 [a] |
| CV (%) | 19.24 | 20.62 | 18.23 |
| Min | 10.05 | 10.09 | 10.20 |
| Max | 25.27 | 29.90 | 24.41 |
| **Length of stomatal pores** | | | |
| Mean ± SD (μm) | 18.80 ± 3.37 [a,b] | 19.56 ± 2.23 [b] | 18.40 ± 2.58 [a] |
| CV (%) | 17.92 | 11.41 | 14.00 |
| Min | 10.41 | 14.52 | 12.15 |
| Max | 27.46 | 25.69 | 27.08 |
| **Width of stomatal pores** | | | |
| Mean ± SD (μm) | 6.97 ± 1.50 [c] | 6.52 ± 1.25 [b] | 5.77 ± 1.12 [a] |
| CV (%) | 21.59 | 19.13 | 19.47 |
| Min | 4.01 | 4.06 | 3.52 |
| Max | 11.94 | 10.30 | 8.63 |
| **Number of stomata per 100,000 μm$^2$** | | | |
| Mean ± SD (μm) | 28.33 ± 4.47 [a] | 27.57 ± 3.53 [a] | 23.07 ± 5.13 [b] |
| CV (%) | 15.79 | 12.81 | 22.25 |
| Min | 21 | 21 | 14 |
| Max | 39 | 36 | 33 |

Ambiguous results were observed regarding the stomatal parameters on both sides of the leaf, where on the upper side, the length of the stomata and width of stomata pores

demonstrated a relatively beneficial effect with nano/macro-ZnPhos in comparison with nano-Hap and NFs free control (Table 4).

On the abaxial side of the leaf, the spray dispersion of nano-Hap proved more effective in terms of stomatal length, stomatal width, and length of stomatal pores in contrast to other variants. However, a statistically significant trend of higher width of stomata pores was observed in NF-free control than nano-Hap and nano/macro-ZnPhos (Table 4).

*3.3. Effectiveness of Spray-Applied Phosphorus Nano-Fertilizers on Quantitative, Nutritional, and Physiological Parameters on Sunflower in 2022 Vegetation Season*

In terms of the number of plants and number of heads, no statistically significant differences were observed between the variants after NFs-treatment per vegetation season (Table 5).

**Table 5.** Comparison of quantitative and nutritional parameters of nano/macro-ZnPhos and nano-Hap-treated variants recorded in 2022 vegetation season.

| Parameter | Control Variant (No Application) | Nano-Hap (Foliar Applied Variant) | Nano/Macro-ZnPhos (Foliar Applied Variant) |
|---|---|---|---|
| Quantitative parameters | | | |
| Number of plants per hectare (pcs) | $77,100 \pm 1000$ | $76,600 \pm 1500$ [ns] | $77,000 \pm 1300$ [ns] |
| Number of heads per hectare (pcs) | $77,600 \pm 1100$ | $76,700 \pm 1400$ [ns] | $77,300 \pm 1000$ [ns] |
| Head diameter (mm) | $178 \pm 22$ | $192 \pm 36$ [ns] | $203 \pm 27$ * |
| Weight of dry head (g) | $88 \pm 34$ | $125 \pm 52$ [ns] | $136 \pm 47$ * |
| Number of seeds per head (pcs) | $1068 \pm 230$ | $1147 \pm 416$ [ns] | $1157 \pm 248$ [ns] |
| Weight of seeds per head (g) | $54 \pm 20$ | $75 \pm 32$ [ns] | $79 \pm 27$ [ns] |
| Seed yield per hectare (t) | $2.95 \pm 0.09$ | $3.04 \pm 0.29$ [ns] | $3.40 \pm 0.06$ * |
| Nutritional parameter | | | |
| Seed oiliness (%) | $42.80 \pm 0.17$ | $43.28 \pm 0.11$ * | $44.45 \pm 0.42$ ** |

The significance: * *p* value < 0.05, ** *p* value < 0.01, [ns] non-significant.

In the context of our results, the nano/macro-ZnPhos variant demonstrated positive statistically significant parameters such as head diameter (mm), dry head weight (g), and seed yield per hectare (t), and it also indicated beneficial quantitative parameters with the number of seed heads (pcs) and weight of seeds per head (g) compared to nano-Hap and NF-free control.

The effectiveness of P-NFs, particularly when utilizing nano-Hap, was found to consist of all observed quantitative parameters; however, the finding did not reach statistical significance, only indicative levels.

With regard to the nutritional variable of oil content, an evident statistical trend was observed, establishing nano/macro-ZnPhos as the most impactful, closely followed by nano-Hap, and finally, the NF-free control.

According to gained dates from seasonal physiology, there is an evident tendency where the most effective response was provided by the spray application of nano/macro-ZnPhos at a statistically significant level for all indices compared to nano-Hap and control (NF-free). A relatively greater trend, but reached on a lower statistical level, was analyzed for nano-Hap and control variants (Table 6).

**Table 6.** List of nano/macro-ZnPhos and nano-Hap-treated sunflower plants' physiological parameters in comparison with NF-free control.

| Parameter | Control Variant (No Application) | Nano-Hap (Foliar Applied Variant) | Nano/Macro-ZnPhos (Foliar Applied Variant) |
|---|---|---|---|
| NDVI [1] | 0.542 ± 0.218 | 0.605 ± 0.155 ** | 0.662 ± 0.041 ** |
| Ig [2] | 1.322 ± 0.597 | 1.503 ± 0.632 * | 1.825 ± 1.002 ** |
| CWSI [3] | 0.450 ± 0.083 | 0.423 ± 0.101 * | 0.390 ± 0.112 ** |

[1] Normalized difference vegetation index (NDVI), [2] Ig Stomatal conductance index (Ig), [3] Crop water stress index (CWSI); The significance: * $p$ value < 0.05, ** $p$ value < 0.01.

Approximately one week after the first NFs application on day 40, a statistical trend of a positive response in the case of nano/macro-ZnPhos and nano-Hap to CWSI and IG indices appeared, which was less noticeable for the NDVI index. Subsequently, the index values stabilized until the second application period. Immediately after the second deposition on day 80, the same trend of positive reactions was recorded for both P-NFs and all physiological indices, reflecting its exceptional tendency until harvest in contrast to the NF-free control (Figure 5).

*3.4. Effect of Spray Deposition of Phosphorous Nano-Fertilizers to Diversity Evaluated on Epigeic Incidence in Real-Field Environment*

Our short-term agro-ecological study showed that under P-NFs-treatment and NF-free control, 3326 individuals of various epigeic terrestrial fauna were collected (Table 7). These individuals were distributed across 17 taxonomical groups with varying spatial distributions among the experimental applications. Among the detected groups, the most abundant populations detected were Coleoptera (29.74%) followed by Collembola (16.51%), Acarina (11.66%) and Formicoidae (11.52%). The dominant groups included Opilionida (7.99%) and Araneida (7.24%). Additionally, Diptera (2.71%), non-determined the stage of Larvae (2.71%), and Orthoptera (2.69%) and Diplopoda (2.49%) were observed as subdominant groups, while Hymenoptera (1.35%) and Heteroptera (1.26%) showed a minor abundance. The remaining five epigeic groups had abundance not exceeding 1%. Generally, the spatial–temporal distribution of collected epigeic populations exhibited a decreasing trend.

From a biodiversity perspective, the most favorable conditions for the life cycle of epigeic groups were observed in the following order: nano/macro-ZnPhos > NF-free control > nano-Hap. The nano-Hap application provided inappropriate conditions for the population's life cycle from all of the variants including the NF-free control.

Similar trend observations were made for the Coleoptera family in foliar deposition with the nano/macro-ZnPhos variant and the NF-free variant compared to the nano-Hap treatment (Table S3, Supplementary Materials). The entire Coleoptera family exhibited the most abundant populations, comprising eight groups, with *Carabidae* showing the highest quantitative dominance at 84.46%. Dominance occurrences were observed also for Staphylinidae, while a subdominant range appeared for Anthicidae and Dermestidae. Abundance levels of other families were lower than 1%.

Furthermore, differences in species abundance were observed between the nano/macro-ZnPhos treatment, NF-free variant, and nano-Hap application within the *Carabidae* family, which encompassed both zoophages and macropterous species. Among these, *Harpalus rufipes* was the most dominant species, predominantly appearing in the late summer and autumn. *Calathus fuscipes* (4.2%), *Brachinus crepitans* (3.82%), and *Poecilus cupreus* (2.51%) with *Pterostichus nigrita* (1.55%) also exhibited higher abundance, while *Pterostichus nigrita* (1.55%) had a minor occurrence. The remaining four species constituted less than 1% of the whole population but contributed to the ambient biodiversity ecosystem (Table S4, Supplementary Materials).

**Table 7.** Abundance and dominance of epigeic groups in the studied treatments with sunflower at the Nitra-Dolná Malanta locality during the 2022 vegetation season.

| Epigeic Group | Control Variant (No Application) | Nano-Hap (Foliar Applied Variant) | Nano/Macro-ZnPhos (Foliar Applied Variant) | Σ | Dominance (%) |
|---|---|---|---|---|---|
| Acarina | 161 | 99 | 128 | 388 | 11.66 |
| Araneida | 64 | 93 | 84 | 241 | 7.24 |
| Coleoptera | 329 | 267 | 395 | 991 | 29.74 |
| Collembola | 199 | 156 | 194 | 549 | 16.51 |
| Dermaptera | 10 | 8 | 10 | 28 | 0.84 |
| Diplopoda | 21 | 39 | 23 | 83 | 2.49 |
| Diptera | 32 | 27 | 31 | 90 | 2.71 |
| Formicoidae | 127 | 94 | 162 | 383 | 11.52 |
| Heteroptera | 18 | 10 | 14 | 42 | 1.26 |
| Hymenoptera | 13 | 17 | 14 | 44 | 1.35 |
| Chilopoda | 6 | 7 | 2 | 15 | 0.45 |
| Isopoda | 15 | 2 | 7 | 24 | 0.72 |
| Larvae | 31 | 26 | 33 | 90 | 2.71 |
| Lumbricida | 1 | - | 2 | 3 | 0.09 |
| Muridae | 1 | - | - | 1 | 0.03 |
| Opilionida | 87 | 59 | 119 | 265 | 7.99 |
| Orthoptera | 20 | 31 | 38 | 89 | 2.69 |
| Σ | 1135 | 935 | 1256 | 3326 | 100 |

## 4. Discussion

### 4.1. Setup of Experimental Base for Conventional Phosphorus Supplied to Soils and Foliar Applied in Form of Nano-Fertilizers of New Generation

Within our locality in Dolná Malanta near Nitra in the Central European Region, the experiment has been conducted on a typical P-deficient Slovak soil per cultivation season [55]. In order to cultivate crops, it is necessary to add an appropriate amount of P to the soil together with other macronutrients (Table 1) annually to reach the legally prescribed limits [56]. Across all variants tested, including the NF-free control, no detrimental effects of P-deficient environment were observed. No evident Fe-uptake limitation resulting in chlorosis or necrosis was observed in relation to the natural development of sunflowers, as detected in a study by Eaton [57].

Thus, its supply was deemed sufficient during the 2022 vegetation season because natural P in hydroxyapatite form inside these soils could be precipitated [30] due to the value of the pH, which in our case is equal or slightly up to the pH > 7 [8]. Additionally, the most likely reduction in P does not occur although hydroxyapatite could control limited solubility, nonbioavailability, and stabilization [41] mainly because the appearance of a free $Ca^{2+}$ cation from feldspar [28] or calcium carbonate [8] was observed in the soil environment. Furthermore, the solubility of similar geochemical forms such as Ca phosphate could be restricted in a neutral or slightly alkaline soil environment [58]. For this reason, the comparatively favorable association with sunflowers in the control (NF-free) variant and P-NFs variants (Table 5), is a rather unexpected finding, particularly considering the presence of certain deviations from the standard climatic norm during months of vegetation (Figure 1, Table S2, Supplementary Materials).

In our short-term experimental trials, we firstly highlighted the concept of an extra-spray deposition of P-NFs but applying a lower dosage to sunflowers due to there being enough P in the soil. Our experimental findings indicated that the application of our novel P-NFs within an optimal spray concentration range did not reflect any statistically significant discrepancy in terms of number of plants and number of heads (Table 5).

### 4.2. Potential Reaction of Phosphorus Nano-Fertilizers Based on Their Physico-Chemical Characteristics

In the case of formed hydroxylapatite (nano-Hap) (Figure S1), it corresponds to a chemically pure phase containing Ca, P and O elements (Figure 3b) with hexagonal symmetry and crystalline size ($L_{vol}$) of 36.4 ± 0.5 nm (Table 2). In terms of morphology, most of the parts are spherical in shape, occasionally with one-direction prolongation with pseudo-hexagonal forms (Figure 3a). This result is relatively surprising regarding its stoichiometric homogeneity as well as its crystalline and size-distribution uniformity because the precursor Sheelavati fish bone came from a heterogeneous nature, even though it was thermally modified at 600 °C.

For the first time, a biocompatible, bioactive, non-cytotoxic NM from biomedical implants [43] was agronomically utilized for crop production via foliar application.

Regarding the chemical and mineral composition of nano-Hap, our results are relatively surprising because no residues were apparent (Figure 3b). Naturally, the chemical composition of the Sheelavati fish bones, before thermal procedures, includes the most typical mineral nutrients such as >1% $CO_3$, Ca, and Mg, or 1% < including Cu, F, Fe, Mn, Si, S, and Zn [36], and the wide range of elements together, in suitable bioavailable forms, should play a significant agronomical role for every plant production as well as sunflower.

Therefore, our hypothesis regarding the formation of nano-hydroxyapatite (nano-Hap) with readily releasable nutrients, owing to its biodegradable origins and preserved fibrous tissue [36], was not confirmed in contrast to the co-precipitation of other P-NFs variants.

The new generation of NFs was synthesized using a co-precipitation method, resulting in a mixture of macro-sized parascholzite and nano-sized calcium zinc phosphate (Table 3). In this context, plant-essential elements such as calcium, phosphorus, carbon, and zinc were identified (Figure 4b). The calcium zinc phosphate shows a crystalline size of 37 ± 3 nm with rhombohedral symmetry and a dominant spherical shape (Figure 4a, Table 3). The parascholzite has a crystalline size ($L_{vol}$) around 130 ± 10 nm, exhibiting monoclinic symmetry and overgrown tabular crystals (Figure 4a).

The solubility and bioavailability of nutrients within the soil solutions or through foliar application can manifest diversely and are influenced by the crystallinity and phosphate structure. As elucidated by Sharma et al. [12], FTIR analysis highlighted mutual bonding within HAP, indicating functional groups ($PO_4^{3-}$, P-O, O-P-O). Upon zinc doping, the calcium position within the polyhedron structure was frequently exchanged due to zinc's incorporation. This zinc substitution in the Hap structure potentially augmented the strength of adhesion forces and introduced distortions to the natural bonding. In the context of our experiment, this led to the gradual release of nutrients in the case of the structurally integrated calcium phosphate and parascholzite in contrast to the pure and structurally more cohesive nano-Hap (Table 2).

Furthermore, considering that both nano-Hap and calcium zinc phosphate exhibit similar sizes (Table 3) and the resultant surface-to-volume ratios, they showcase analogous solubility and chemical reactivity. Meanwhile, parascholzite, functioning as a micro-fertilizer, exhibits a slower nutrient release rate attributed to its larger size and partitioned zinc content.

### 4.3. Reaction of Spray-Dispersed Phosphorus Nano-Fertilizers with Sunflower Leaf Structure—Stomata

A lack of P reduces stomatal density and decreases the stomatal size and aperture, which can cause a decrease in stomatal conductivity, $CO_2$ absorption and a reduction in the rate of photosynthesis [59]. When it comes to the application of fertilizers at the nanoscale, we anticipate a higher agronomic effect as compared to bulk fertilizers with chemically similar forms. This phenomenon has been elucidated in a study by Xiong et al. [30], which also examined the implication of natural hydroxyapatite to the soil system.

Fernández et al. [60] investigated the effect of different doses of P in soils on selected leaf properties for wheat (*Triticum aestivum* cv. Axe). The various impacts on the leaves were evidenced at the highest P concentration in soil at 24 kg.ha$^{-1}$. Obviously, the highest numbers of stomata and trichomes were recorded, particularly on the adaxial side. Furthermore, they found that the total leaf area, cuticle thickness, and contact angle decrease with P deficiency while the work-of-adhesion increases with water application. Additionally, under P deficiency, the absorption of P spray deposition was partly restricted, which was likely due to a reduction in leaf permeability.

In our proof-of-concept, ambiguous results were observed, as the highest number of stomata was recorded on both the adaxial and abaxial sides of the sunflower leaves in the NF-free control variant. Surprisingly, the application of both phosphate-based NFs, except the width of the pores of the stomata, resulted in a slightly more efficient stomatal parameter (Table 4). This trend may be attributed to the manner in which phosphate deposition during spraying and the direct action of P influence stomatal characteristics rather than its translocation from the soil environment.

One of the pathways for the NPs to enter the plant is through the leaf stomata [61]. In the case of sunflower, the average stomata diameter corresponds to 28–30 μm [28]. Once they have interacted with the leaf surface, cell-to-cell transport within the plant could be worked on through plasmodesmata. Plasmodesmata have channels with a size range of 50–60 nm that are capable of communicating within cells and transporters [61], where both of our applied P-NFs had a size lower than 50 nm (Tables 2 and 3).

In terms of the reactivity of nano-Hap and nano/macro-ZnPhos on the leaf surface, we assume that there was less generation of reactive oxygen species (ROS) compared to, for instance, more photoactive $TiO_2$ NPs [62]. Due to this, we hypothesize that P-NFs, especially nano-Hap, could be more biocompatible, bioactive, less cytotoxic and thus more absorbable to leaf cuticula or other leaf structures.

Ultimately, the augmentation of effects observed upon the application of nanoscale fertilizers in contrast to traditional agronomic fertilizers is expected to enhance leaf permeability. This, in turn, fortifies leaf tissue and contributes to an improved mineral nutrient foundation. Consequently, this accelerated growth directly influences production-related parameters, as Mahmoodi et al. [63] observed with *Borago officinalis* L. and Elsayed et al. [64] observed in the *Rosmarinus officinalis* plant.

### 4.4. Effects of Nano-Hap and Nano/Macro-ZnPhos Application on Selected Quantitative, Physiological, and Nutritional Indicators in Sunflower during 2022 Vegetation Season in Central European Region

The results showed that the head diameter, head weight, and seed yield per hectare were significantly higher with the spray-dispersion of phosphorus-containing variants than without it (Table 5). This is not surprising considering that P is one of the essential nutrients for plant growth and development, including energy acceleration in plants [15,17], regulation of water uptake and root system development [14], and flower and fruit production, ultimately stimulating sunflower yield and yield-forming parameters [12,13].

It could be said that our spray-dispersed P-NFs confirmed the suitable agronomic strategy at low concentrations corresponding to 33 mg·L$^{-1}$.

Evidently, the substantial increase in sunflower's fresh biomass was successfully demonstrated through the utilization of three surface-modified nano-Hap variations, in contrast to conventional triple-super phosphate and rock-phosphate, within an acidic soil environment [30]. Conversely, this success was not observed within an alkaline soil environment (pH = 8.2). It is likely that predetermined P immobilization played a role in the latter scenario. Nonetheless, in our study, the spray-application approach significantly accelerated the differentiation in quantitative parameters such as head diameter, head weight, seed yield per hectare, and sunflower oil content when compared to the NF-free control.

Similarly, Mishra et al. [61] found increases in various growth-promoting parameters including shoot and root length, shoot and dry weight, shoot and root fresh weight, and

leaf area when applying CaP-urea with particle size > 100 nm through spray deposition at a low concentration of NFs on a finger millet under drought and irrigated conditions. Also, a statistically significant difference was observed with a grain yield of wheat of more than 20% when zinc-and magnesium-doped hydroxyapatite NPs modified with urea were utilized [12], which fully agrees with our results.

An increase in the yield of seeded rice (*Oryza sativa* L.) was ensured via foliar nutrition application (five times) in the form of ionically soluble multi-component fertilizers, including phosphorus and zinc, as reported by Shaygany et al. [65]. However, it should be noted that in our case, we anticipate a slightly different nano-domain effect applied only two times more than the many-times dispersion of pure soluble species of P-NFs on sunflower or other crops. So, the NFs application may have a larger effect than the application of ionic-soluble counterparts at the same concentration.

Regarding the other quantitative parameters, such as the number of seeds and weight of seeds per head, no statistically significant differences were observed between variants (Table 5). In this regard, our one-year results are relatively comparable to the research conducted by Sawan et al. [66], who studied the combination of spray deposition of P and Zn fertilizers on Egyptian cotton (*Gossypium barbadense* L.). The application encouraged dry matter yield, an increase in the number of open balls, ball weight seed cotton yield per plant, and seed cotton and lint yield per hectare. P could play a positively role in the flowering and floral development, particularly in the context of disc-shaped flowers [67], which is in good agreement with our number of seeds per head within the applied P-NFs, albeit primarily serving as an statistically indicative level (Table 5).

Moreover, the slight differences among P-NFs variants could be an aspect of more effective uptake of nitrogen from soils, which was supported with conventional NPK (Table 1). This is a crucial factor in crop development, and it might subsequently impact the insect pollination ability, where a relatively higher number of bumblebees compared to bees in control variant was present. Other factors influencing the crop yield, such as local climatic fluctuations (Figure 1) as well as agrotechnical processes that restricted weed manifestation, remained comparable in all experimental variants.

In the case of the application of a multi-nutrient complex containing zinc and magnesium, nano-Hap, and urea integrated to soil, we expected the qualitative and quantitative parameters to intensify, and there was a positive effect on the physiological phospholipid levels in wheat [13].

Based on our sunflower one-season physiology (Table 5), similar to the evaluation of quantitative parameters, it is evident that the most effective treatments were again the P-NFs. In these cases, it could be assumed that the effect of P was further enhanced by an increased nitrogen content in the plant, particularly in the leaves, which has a close relationship with stomatal regulation and movement [61] as well as stomatal conductance. This was directly confirmed with the Ig physiological index (Table 5) and indirectly through the variability of sunflower leaf stomatal parameters (Table 4).

Furthermore, it is expected that leaves with a higher nitrogen content also indicates higher chloroplast content [61] or potentially a more intense impact on photosynthesis [68].

Even in the evaluation of CWSI, it is evident that the lowest stress response to potential water deficit confirmed the mentioned conclusions for both P-NFs treatments.

Increased chlorophyll *a* and *b* values, more than twofold, along with other biochemical parameters, were observed with the spray application of macro-sized CaP with urea on finger millet [61]. Jacob et al. [69] found that a reduction in soil inorganic phosphorous resulted in a decrease in total chlorophyll, specific leaf weight with higher water content, and total soluble protein. They also verify that P deficiency caused a reduction in the light-saturated photosynthetic rates, $CO_2$ assimilation quantum yield, and carboxylation efficiency in sunflowers. Another study showed a reduction in the photosynthetic activity of photosystem II (PSII) during growth under P deficiency in maize and sunflower leaves [70].

In our short-term experiment trials, we observed an indication of positive response from both P-NFs, particularly in the early stages after the initial application focused on the

stage of leaf development, and subsequent spray deposition during stem elongation and flower bud formation (Figure 5), which continued until harvest.

However, in comparison with both P-NFs, it must be stated that nano/macro-ZnPhos was more effective than nano-Hap. Although zinc was present at low concentrations, it made a substantial difference. In the case of sunflowers, zinc is responsible for various enzymatic processes such as protein synthesis, hormone regulation, and amino acid formation. It is an integral component of many proteins, and it is also associated with photosynthesis and chlorophyl content [27,71,72]. Jan et al. [27] reported improved physiological indicators of sunflowers under drought stress with the application of the conventional exogenous agronomical zinc form ($ZnSO_4$). This intensified the chlorophyll content, both *a* and *b*, and even accelerated the photosynthetic rate by around 40%, also with significant antioxidant enzyme activity. Regarding the combination of soil-applied potassium with spray dispersion of Zn and P on Egyptian cotton, a beneficial effect on total chlorophyll appeared [66].

In our previous studies, ZnO NPs foliar application was used on several crops, including foxtail millet [73], lentil [74], and sunflower [8]. A favorable response across all monitored indices, including NDVI, Ig, and CWSI, was evident in terms of seasonal physiology, much like what was observed in sunflower throughout the 2022 vegetation season. Consequently, in terms of physiological progress induced by ZnO application, noteworthy improvements were detected approximately one month after the spray dispersions, as manifested by enhancements across all physiological indices. Notably, this timeline mildly coincided with the nearly immediate favorable responses elicited by both P-NFs (Figure 5).

### 4.5. Evaluation of Terrestrial Insect Diversity following Application of Phosphorous Nano-Fertilizers at Sunflower Cultivated Fields

The distribution of insect communities plays a crucial role in biodiversity and ecosystems in agriculture, where insects play a major role in the interaction between plants, animals, and the ambient environment. Also, in our short-term study, their abundance reflects the processes associated with application of P-NFs, where other factors such as agrotechnical management, geological and other natural settings, and local climatic fluctuations were the same during the experimental procedures. The only real difference was the application of two slightly different chemical compositions of P-NFs, where exposure to P and Ca was relevant in the case of nano-Hap (Figure 3b) and insignificant concentration ranges of Zn and C were additionally utilized in nano/macro-ZnPhos (Figure 5b), which probably encouraged the series of changes on the incidence of epigeic populations (Tables 7, S3 and S4).

Maintaining complex and diverse ecosystems is an essential strategy for balancing populations within integrated communities [75].

Our results, based on collected epigeic populations (Tables 7, S3 and S4), showed a non-uniform trend. As expected, the nano/macro-ZnPhos treatment and the NF-free control had a wider bio/diversity of epigeic individuals, while nano-Hap application showed a relatively negative impact on biodiversity. This is one of the most surprising facts because apatite is a naturally dominant mineral in the soil environment with extensive agronomic applications [30]. This outcome implies a noteworthy agronomic aspect associated with the application of more intricate phosphorus-based fertilizers encompassing zinc through spray dispersion onto sunflower crops. On the contrary, it is worth assessing the insecticidal potential of nano-Hap. In this regard, the nano-Hap variant displayed the lowest population of terrestrial insects while concurrently maintaining relatively higher sunflower yields and yield-related parameters. This outcome aligns with findings observed with MgO-NPs and cowpea (*Vigna unguiculata* L.) [3]. In this context, NPs have the potential to penetrate an insect's circulatory system, inducing toxic effects on the lymphatic systems, ultimately leading to the generation of oxidative stress and mortality [76].

The outcomes of our one-year study underscore a noteworthy occurrence with the dominant family *Carabidae* constituting a significant proportion (84.46%). This prominence is attributed to its positive response to the applied agronomic practices, ambient vegetation, microclimate conditions, and trophic preferences. Moreover, our hypothesis concerning the low-level P dosage gains significance as it serves to mitigate the risks associated with eutrophication and counteracts negative side effects within the natural phosphorus–agronomic cycle. Our results distinctly reveal that the most favorable variant corresponds to nano/macro-ZnPhos, which is followed by the NF-free control and pure nano-Hap. In a related context, Mhlanga et al. [77] identify the correlation between soil moisture, the pH, initial nitrogen application, and P content and the variability of ground insect diversity dynamics, suggesting their use as suitable indicators for agroecological development and changes.

Throughout the vegetation period, a total of eight species were identified within the *Carabidae* family. In terms of noteworthy observations, the prevalence of the ubiquitous and eudominant field species *Pseudoophonus rufipes* (86.98%) holds significance. Its high abundance serves as a clear indication of its adaptation to anthropogenic influences underscored by its prominence [78], which is also supported by our results (Table S4). Its maximum occurrence was recorded in late summer and autumn in the nano/macro-ZnPhos in contradiction to other variants. This species is able to survive overwinter in the larval stage and as an adult contributes to the regulation of aphid populations or other microbiomes during the vegetation period and thus influencing farm management systems appropriately [78]. On the other side, it is also a granivorous species, consuming weed and medical plants seeds.

## 5. Conclusions

In this one-season agronomical study, we have employed biocompatible and bioactive nano-Hap, originally developed for biomedical purposes, for foliar application on sunflower under real field conditions. Furthermore, we conducted a comparative analysis with another promising P-NF composed of a combination of nano-sized calcium zinc phosphate and macro-sized parascholzite (nano/macro-ZnPhos), which was synthesized through a chemical co-precipitation method. The impact of these innovations was explored in the field conditions of central Europe during the 2022 vegetation season, employing sunflower (*Helianthus annuus* L.) as the model crop and juxtaposing against a nanofertilizer-free control (NF-free). Almost instantly, both phosphate nanofertilizers exhibited noteworthy agronomic advantages, as evidenced by enhanced crop yields, improvements in yield-related parameters, and an increased nutritional oil content. Among these, the most statistically significant positive response was indicated in the nano/macro-ZnPhos-treated variant followed by the nano-Hap-treatment and, finally, the control variant.

A similar positive trend was also shown when analyzing the overall seasonal physiological parameters, including NDVI, Ig and CWSI. However, it should be noted that in field experiments, the ideal response was demonstrated around one week after the initial and subsequent NFs leaf dispersion, with the trend persisting until harvest.

Delving into the diversity of terrestrial insect populations within the agricultural ecosystem, the nano/macro-ZnPhos variant again exhibited the most favorable conditions. The more agronomically studied nano-Hap showcased the lowest biodiversity.

Our brief results confirm the relevance of using low-dosage spray-dispersed P-NFs, supporting the concept of precision and sustainable agriculture under field conditions without any negative environmental impacts observed. The study was conducted over a year and the apparent initial agronomical tendencies will require validation in subsequent growing seasons.

**Supplementary Materials:** The following supporting information can be downloaded at: https://www.mdpi.com/article/10.3390/agronomy13102606/s1, Figure S1: X-ray diffraction powder patterns of hydroxylapatite formed from Sheelavati fish bone; Figure S2: X-ray diffraction powder patterns the mixture of parascholzite and calcium zinc phosphate (nano/macro-ZnPhos) formed by chemical co-precipitation method; Table S1: Basic values of electrical conductivity and the pH of applied nano-fertilizers compared to applied water as a control variant; Table S2: Comparison of monthly temperature and precipitation characteristics during the vegetation season of 2022 with the long-term norm from 1991 to 2020; Table S3: Abundance and dominance of the Coleoptera family in the studied treatments with sunflower at the Nitra-Dolná Malanta locality during vegetation season 2022. Table S4: Abundance and dominance of the *Carabidae* species in the studied treatments with sunflower at the Nitra-Dolná Malanta locality during vegetation season of 2022.

**Author Contributions:** Investigation, writing—original draft preparation, supervision, proposed the topic, D.E. and M.K. (Marek Kolenčík); obtained formal analysis, figures, tables, and software—statistical implementations, and validation, R.I., V.Ž.Č., I.Č. and S.G.R.; writing—review and editing, and supervision, S.S. and Y.Q.; writing—review and editing, checked grammar, review and editing whole concept of manuscript, M.Š.; performed analysis of oil, I.Č.; formed phosphate-nano-fertilizers, B.R.S.; obtained and interpreted agroecological systems, J.I.P. and M.B.; formed and interpreted the X-ray data analysis, E.D.; performed microscopic investigation of sunflower leaf surface, Ľ.Ď., S.K., L.T., N.K. and M.K. (Mária Kalúzová); performed SEM and EDS analysis, G.K.; funding acquisition and project administration, D.E., M.K. (Marek Kolenčík) and L.D. All authors have read and agreed to the published version of the manuscript.

**Funding:** This research was funded by the Grant Agency of the Slovak Republic Ministry of Education and the Slovak Academy of Sciences under contract VEGA 1/0655/23, VEGA 1/0359/22 and by the European Union foundation (Erasmus Plus Programme for academic staff mobility) and postgraduate program sponsored by the National Scholarship Programme of the Slovak Republic trough SAIA Organization.

**Data Availability Statement:** Not applicable.

**Conflicts of Interest:** The authors declare no conflict of interest.

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
