# Peer review of "Significance of Phosphate Nano-Fertilizers Foliar Application: A Brief Real-Field Study of Quantitative, Physiological Parameters, and Agro-Ecological Diversity in Sunflower"

_agronomy, doi:10.3390/agronomy13102606_

Round 1

Reviewer 1 Report

-          The current manuscript "Significance of Phosphate Nano/fertilizers Foliar Application: Real-Field Study of Quantitative, Physiological Parameters, and Agro-Ecological Diversity in Sunflower". The idea seems to be good in the application, but the characterization of the NFs requires further clarification.

Comments:

-          The abstract part needs to rewrite in a way to define the exact novelty and originality of your work.

-          All abbreviations used should be mentioned in the place of their first mention followed by an abbreviation and then only the abbreviation is written.

-          Keywords: quantitative and physiological parameters?

-          The characterization of NFs are incomplete and requires the following analyses to be carried out: TEM, Raman, DLS, and XPS analysis.

-          The introduction must be completed by clarifying the main objectives of the research and by motivating the experimental strategy adopted by authors.

-          In the introduction there are some paragraphs without references.

-          Authors should add these recent references at the end of the sentence, lines 55-57 "Applied NFs follow a definition similar to that of nanomaterials, where one of the three dimensions should not exceed 100 nm; otherwise, it becomes a mixture of NF with micro-sized fertilisers" https://doi.org/10.1016/B978-0-323-91933-3.00010-6; https://doi.org/10.1002/biot.202300301; https://doi.org/10.1007/978-981-99-2419-6_12

-          "fertilisers" or "fertilizers"

-          In introduction section, author should be clarify the advantages of synthesis nanoparticles and give more knowledge about their application. I recommended the following reference for citing: https://doi.org/10.1007/s12011-020-02138-3; https://doi.org/10.1007/s00203-023-03467-2

-          Add all materials used in Materials section.

-          Add references to all methods.

-          The authors should add more explanations to the results.

-          The related studies and deep discussion regarding to the all parts in the manuscript.

-          The whole manuscript must be checked to avoid the presentation of same information several times.

-          Conclusion: I think author should try to link better their work.

-          The English language needs to be significantly improved, in wording, grammar and sentence structure.

-          I suggest the authors to go through the manuscript one more time to minimize some errors, typos etc.

-          The English language needs to be significantly improved, in wording, grammar and sentence structure.

Author Response

Reviewer 1

# Comment 1 The abstract part needs to rewrite in a way to define the exact novelty and originality of your work.

* Answer 1: We are in agreement with the aforementioned assertion. (L38-42) …“The empirical evidence presented herein indicates that the utilization of biocompatible and bioactive nano-Hap, initially engineered for biomedical applications, and nano/macro-ZnPhos, now foliarly applied at reduced concentrations, elicited a statistically significant elevation in quantitative parameters and seasonal physiological responses.”…

# Comment 2 All abbreviations used should be mentioned in the place of their first mention followed by an abbreviation and then only the abbreviation is written.

* Answer 2: Certainly, we concur that all abbreviations have been revised in accordance with the reviewer’s recommendations.

# Comment 3 Keywords: quantitative and physiological parameters?

* Answer 3: The keywords were modified in accordance with reviewer suggestion. The following corrections were applied (L51) …“quantitative parameters; physiological parameters”…

# Comment 4 The characterization of NFs are incomplete and requires the following analyses to be carried out: TEM, Raman, DLS, and XPS analysis.

* Answer 4: In the context of our manuscript, we have adopted a chemical verification approach to evaluate nanofertilizers. This entails the application of energy-dispersive X-ray spectroscopy (EDX), as depicted in both Figure 3b and Figure 4b. Furthermore, we have complemented this method with crystallographic determination, as visually represented in Figure S1, S2. To guarantee precision, we have determined detail crystallographic structure parameters using the TOPAS software program, and the results are comprehensively presented in Table 2, Table 3.

Through the recommended methods of RAMAN and XPS analysis it could be analyzed only the mutual functional groups and certain bonds, which, however, we have already determined through a detailed X-ray diffraction record, in conjunction with verification through EDS.

We appreciate the recommendations for these selected methods, which are better suited for the determination of complex matrices with many inorganic and organic components. However, this is not our case, as we have produced and utilized crystalline phases with high-temperature treatment which reflected good stoichiometry and crystallinity. The SEM instrument used possesses a similar limit of detection as TEM. Therefore, we have chosen to retain these results without further TEM analyses.

We concur that, for the determination of NFs using DLS application, it is crucial to be acquainted with specific characteristics, such as the ionic strength of the solution - conductivity, which we have incorporated into the methodological section of the manuscript. Additionally, we have assessed the acidity-basicity, i.e., the pH of the applied dispersion solution, which is also included in the manuscript under (L170-L174) …“…The electrical conductivity of the applied nano-fertilizer samples as well as the control variant was measured using the conductivity EC meter DiST5 (HANNA, RI, USA), and the pH values were analyzed using pH meter inoLab 730 (Weilheim, Germany). The values are presented in Table”…

# Comment 5 The introduction must be completed by clarifying the main objectives of the research and by motivating the experimental strategy adopted by authors.

* Answer 5: We respectfully disagree with the assertion. In the introduction, several key concepts addressing the main objectives directly related to the essence of the article and individual outcomes have been elaborated, such as i) the advantages of applying nanofertilizers (L60) “Generally, the size of nanoparticles (NPs) in NFs, compared to their analogical chemical macro-sized or ionic counterparts, implicates different reactivity, mobility, and bioavailability”, or ii) the rationale behind selecting sunflowers (L66) “…It has wide leaves with a surface morphology that makes it ideal for spray deposition of NFs…”, iii) the novel types of fertilizers in the combination of nano and macro fertilizers (L79) “…have indicated research involving nano-phosphorus combined with other nutrients, the broader interaction between diverse NF categories and their agronomical impacts remains relatively unexplored…”, iv) the creation of new fertilizer types based on the use of secondary food raw materials (L119) “…the utilization of secondary food waste derived from animal sources to produce HAP carries significant advantages…”, and so forth. Additionally, currently concept of the article does not support the detailed description of experimental strategies in the intro-section, and suggesting concept of adopted experimental strategy we prefer apply in a review or a methodologically-type of publication.

# Comment 6 In the introduction there are some paragraphs without references & Authors should add these recent references at the end of the sentence, lines 55-57 „Applied NFs follow a definition similar to that of nanomaterials, where one of the three dimensions should not exceed 100 nm; otherwise, it becomes a mixture of NF with micro-sized fertilisers“ https://doi.org/10.1016/B978-0-323-91933-3.00010-6; https://doi.org/10.1002/biot.202300301; https://doi.org/10.1007/978-981-99-2419-6_12 & In introduction section, author should be clarify the advantages of synthesis nanoparticles and give more knowledge about their application. I recommended the following reference for citing: https://doi.org/10.1007/s12011-020-02138-3; https://doi.org/10.1007/s00203-023-03467-2

* Answer: Thank you for the recommendation, the all six citations were supplemented based on the reviewer’s recommendation.

# Comment 7 „fertilisers" or "fertilizers“

* Answer: We concur; it has been standardized in the manuscript.

# Comment 8 Add all materials used in Materials section & Add references to all methods.

* Answer: Since this research constitutes an original contribution, citations may appear redundant in certain cases. Nonetheless, we acknowledge the absence of citations in specific instances, notably in the section elucidating the hybrid sunflower, where we have subsequently incorporated it “…high resistance against Sclerotinia sclerotiorum and Diaporte helianthi, while also displaying tolerance to Plasmopara halstedii [44]…”.

# Comment 9 The authors should add more explanations to the results.

* Answer: In essence, we do not intend to expand the results section with further elaboration, as we consider the description of the outcomes to be sufficient, a point that Reviewer 4 also emphasized “…your results section is exceptionally well-written and presents intriguing findings…”.

# Comment 10 The related studies and deep discussion regarding to the all parts in the manuscript.

* Answer: We agree, the discussion was revised in accordance with the opponent’s proposal.

# Comment 11 The whole manuscript must be checked to avoid the presentation of same information several times.

* Answer: We agree, the similar-sound information in the several parts of manuscript were omitted.

 # Comment 12 Conclusion: I think author should try to link better their work.

* Answer: We agree, the text has been revised at several places in the conclusion for better connection, e.g. (L730 - 750) “In this study, we have employed biocompatible and bioactive nano-Hap, originally developed for biomedical purposes, for foliar application on sunflower under real field conditions. Furthermore, we conducted a comparative analysis with another promising P-NFs, composed of a combination of nano-sized calcium zinc phosphate and macro-sized parascholzite (nano/macro-ZnPhos), synthesized through a chemical co-precipitation method.” or “Almost instantly, both phosphate nanofertilizers exhibited noteworthy agronomic advantages, as evidenced by enhanced crop yields, improvements in yield-related parameters, and an increased nutritional oil content. Among these, the most statistically significant positive response was observed in the nano/macro-ZnPhos-treated variant, followed by the nano-Hap-treatment, and finally, the control variant.”.

# Comment 13 The English language needs to be significantly improved, in wording, grammar and sentence structure & I suggest the authors to go through the manuscript one more time to minimize some errors, typos etc.

* Answer: The manuscript was completely checked for grammatical mistakes and typos.

Reviewer 2 Report

The aims of this study highlight impact of foliar spray deposition of two types of phosphorus-based nano/fertilizers to quantitative indices, physiology of sunflower (Helianthus annuus L.), and terrestric agro-ecological population under field conditions at vegetation season 2022 in Central European Region.  Two sources on nano/fertilizers (NFs) were compared with a NF free control. Two applications were done during the crop cycle. Crop variables were measured (yield, gain number, number of heads, leaf stomata, NDVI, CWSI,  etc) and  terrestric agro-ecological population were measured.

The main is related to the fact that the results and conclusion are supported by only one field experiment conducted in only one season and one site. So the reliability of the study and its conclusion is weak.

The rationales of crop measurements such as NDVI, Ig, CWSI and terrestrial agroecological population are not presented in the state of the art.

In addition, the statistical analysis applied is confusing. See comments in the pdf file.

more comments are presented in the pdf file attached.

Author Response

Reviewer 2

# Comment 14 The authors emphasize the pioneering character of research related to the use of NFs. The advantage of the conducted research is the wide range of analysed parameters. However, the shortcoming is only one-year field trials. For this type of research, they should be conducted in a cycle of at least 2 years. One-year results showed that both phosphorus NFs demonstrated remarkable agronomic benefits, manifesting in improved yields, yield-forming parameters, and enhanced nutritional oil content.

* Answer: Thank you for appreciating the benefits of our research. Regarding the one-year field trials in our case, it is not possible to repeat the experiment on the same experimental site, and conditions due to crop rotation system. We agree with the idea of a two-year experiment, however, for gradual, often radical climate changes with strong local fluctuate conditions also confirmed with Guo et al. (2022), and Seppelt et al. (2022) it is not possible, we decided to present only one-year experiment, which cannot be replicated precisely twice with equivalent conditions. Once again thank you for your suggestion, and for the next sets of experiments we should evaluated it. Also, we think that nanoparticles-treatment could be strong tools for fight with gradual climate changes but unfortunately results will not be able to reproduce for next years, or interpretable due to local weather conditions.

# Comment 15 Weather conditions should be discussed against the background of precipitation and temperature over long-term period

* Answer: Thank you for your suggestion. We have added a statement to the assessment of weather conditions in terms of the long-term climatic normal for the experimental locality “this finding is rather unexpected, particularly considering the presence of certain deviations from the standard climatic norm during months of vegetation (Figure 1, Table S2, Supplementary Materials).” with table in supplementary section “Table S2”.

# Comment 16 In table 7-9, the number of species from three variants was summed up and the dominance of individual groups was calculated; why dominance was not calculated separately for each of the three objects of the experiment.

* Answer: We agree, the tables have been adjusted according to the reviewer’s suggestion.

# Comment 17 The authors did not justify the advisability of studying the species diversity of the insect community associated with the application of phosphorus-NFs; this chapter in the discussion should be supported by relevant literature.

* Answer: In the context of agro/biodiversity and phosphorus nanoparticles, there is a notable absence of relevant academic literature. Our intent was to offer potential readers a genuine insight into the occurrence of edaphic groups in the Central European region, which we consider to be of significance. Furthermore, this serves as one of the novel aspects and value additions that we highlight both in the abstract and in the conclusion of the manuscript, with a certain citation potential.

Reviewer 3 Report

The authors emphasize the pioneering character of research related to the use of NFs. The advantage of the conducted research is the wide range of analyzed parameters. However, the shortcoming is only one-year field trials. For this type of research, they should be conducted in a cycle of at least 2 years. One-year results showed that both phosphorus NFs demonstrated remarkable agronomic benefits, manifesting in improved yields, yield-forming parameters, and enhanced nutritional oil content.

Comments:

- weather conditions should be discussed against the background of precipitation and temperature over long-term period

- in table 7-9, the number of species from three variants was summed up and the dominance of individual groups was calculated; why dominance was not calculated separately for each of the three objects of the experiment

- the authors did not justify the advisability of studying the species diversity of the insect community associated with the application of phosphorus-NFs; this chapter in the discussion should be supported by relevant literature.

Author Response

Reviewer 3

# Comment 18 Please let me know why the authors use the term nano/fertilizers with a slash. In the studied literature, the term nano-fertilizers or simply nanofertilizers is often used. & Please change nano/fertilizers to nano-fertilizers throughout the manuscript.

* Answer: We agree, “nano/fertilizers” was changed to nano-fertilizers throughout the whole manuscript including the title of manuscript “Significance of Phosphate Nano-Fertilizers Foliar Application: Real-Field Study of Quantitative, Physiological Parameters, and Agro-Ecological Diversity in Sunflower”

# Comment 19 Please check the name hydroxylapatite. It should be hydroxyapatite. Is it correct? Please confirm.

* Answer: The term of mineral “hydroxylapatite” corresponds to nomenclature and was employed accurately. Also, hydroxylapatite was confirmed through X-ray analysis, as depicted in the attached data of the analysis, and also in supplementary data (Fig. XY). Furthermore, we have included a brief note in the manuscript, highlighting that, in conjunction with other minerals, it belongs to a crystalline modification of apatite (L97-L100) “A notable example is apatite, the most common naturally occurring compound containing P with various crystal modifications such as hydroxylapatite, chlorapatite and fluorapatite.

Fig. 1. The original X-ray record corresponds to hydroxylapatite.

# Comment 20 Before using the abbreviation please explain means: nano fertilizers (NFs)

* Answer: The abbreviation NFs was already explained in manuscript (L54): “The progress in nanotechnology is beginning to integrate fields such as agriculture, where the development of nano-fertilizers (NFs) is one of the most significant advancements.”, and their properties was also described (L60) “…Generally, the size of nanoparticles (NPs) in NFs, compared to their analogical chemical macro-sized or ionic counterparts, implicates different reactivity, mobility, and bioavailability…”.

# Comment 21 Before using the abbreviation please explain means: hydroxyapatite (HAP)

* Answer: Please, check # Comment 19.

# Comment 22 Please add subsection: 2.1.1. Formation of hydroxyapatite & Please add subsection: 2.1.2. Formation of mixture nanosized calcium zinc phosphate and macro-sized parascholzite

* Answer: Both subchapter titles were added according to the opponent's suggestion (L153) “2.1.1. Formation of Hydroxylapatite” and (L161) “2.1.2. Formation of Mixture Nano-Sized Calcium Zinc Phosphate and Macro-Sized Parascholzite”.

# Comment 22 Please add equipment producer as follow (Name, City, State, Country) (Bruker, Billerica, MA, USA).

* Answer: The equipment producers were added in accordance with reviewer suggestion (L170) “JEOL, Akishima, Tokyo, Japan” and (L175) “Bruker, Billerica, MA, USA”.

# Comment 23 MDPI journals strongly recommend the use of SI units, with the exception of 1 ha which is accepted as a worldwide unit of area. Please change L to dm3 & Please add (Producer/manufacturer, City, Country) & Please use abbrev. CA & (CLAAS GmbH & Co.,Harsewinkel, Germany) please confirm. & 287: dm3 & Please center text in rows: Quantitative parameters and Nutritional parameter

* Answer: All mentioned suggestions were accepted, the units were corrected according to SI system “concentration 4 dm3·ha-1, and Pictor® fungicide at dose 0.4 dm3 ha-1, and concentration 33 mg·dm-3, respectively.

# Comment 24 There is no clear definition of treatments in the methodology. Authors must write how the individual objects of the experiment were treated. 15-15-15 DUSLOFERT fertilizer was applied to all objects, but in what dose? 3 series of experiment were used: 1) without nano-fertilizers, 2) nano-hydroxyapatite (nano-HAp) 3) nano-calcium zinc phosphate + macro-sized parascholzite (nano/macro-ZnPhos). Please explain it clearly in the methodology.

* Answer: We agree, that was added to the manuscript (L228) “…The 15-15-15 DUSLOFERT fertilizer (Duslo, a. s., Šaľa, Slovak Republic) was dosed at 400 kg.ha-1 through machinery…” and information about P-NFs concentration has been already found in manuscript (L238) “Dispersion with phosphorus-based NFs were applied both at concentration 33 mg·dm-3.

# Comment 25 Please specify for what purpose the ultrasonication was carried out?

* Answer: We concur, within the text, a brief explanation was integrated. (L245) “Prior to each foliar deposition to sunflower, the NFs samples were subjected to a 15-minutes ultrasonic treatment for better colloidal-dispersion properties.

# Comment 26 Days after sowing or days after emergence? Did the authors plant sunflowers?

* Answer: Surely, the correct term is “sowing” which was add and modify in many different places of manuscript.

# Comment 27 Please check the syntax error of this sentence.

* Answer: The syntax in the sentence was corrected “To accurately assess changes in stomata distribution and their fundamental characteristics, a leaf area of 1mm2 (equivalent to 1,000,000 µm2) was estimated.”

# Comment 28 Please check again. See line 362 and comments to table 4a and 4b. Should it be 100,000 μm2?

* Answer: The number adjustments were made in Table 4a and 4b according to reviewer’s suggestion.

# Comment 29 megagrams per hectare (Mg ha-1).

* Answer: Thank you for your recommendation. However, we have decided not to accept your comment. In the past, we published an article in MDPI-Plants, where they accepted the t ha-1 (https://www.mdpi.com/2223-7747/12/9/1789). Within MDPI-Agronomy, the entered unit is also accepted (https://www.mdpi.com/2073-4395/13/9/2349).

# Comment 30 Figure 4b. Please add a description of the Y-axis & Figure 5b. Please add a description of the Y-axis

* Answer: We concur that the Y-axis was supplemented in accordance with the opponent’s proposal.

# Comment 31 Please change Table 4a title. Shouldn’t the entries marked with red color be in the footer below the table?

* Answer: We have opted to retain the data in the table’s title.

# Comment 32 Why did the authors give the number of stomata per 100,000 μm2 and per 1 cm2? Isn’t it enough to specify per 100,000 μm2?Table 4a and 4b I suggest removing this table row Number of stomata per 1 mm2

* Answer: Table row Number of stomata per 1 mm2 was removing in accordance with reviewer suggestion.

# Comment 33 Please consider deleting this part of the manuscript. The results described earlier are sufficient for this publication. I propose to use the results on agrobiodiversity in another publication on the impact of nanofertlizers on the species composition of various epigeic terrestrial fauna. & Please see the comments (rows 410, 667).

* Answer: In the context of conducting “agroecological” analyses in the Central European region, we endeavor, and will continue to do so in the future, to engage in a more comprehensive research approach focusing on pertinent agronomic and ecological parameters. Consequently, we disagree with the opponent regarding the omission of certain analyses and their interpretation in the relevant sections of the manuscript.

Reviewer 4 Report

Review Report

Manuscript ID: agronomy-2599185

Title: Real-Field Study of Quantitative, Physiological Parameters, and Agro-Ecological Diversity in Sunflower

Authors: Dávid Ernst, Marek Kolenčík, Martin Šebesta, Ľuba Ďurišová, Samuel Kšiňan, Lenka Tomovičová, Nikola Kotlárová, Mária Kalúzová, Ivan Černý, Gabriela Kratošová, Veronika Žitniak Čurná, Jana Ivanič Porhajašová, Mária Babošová, Edmund Dobročka, Yu Qian, Sasikumar Swamiappan, Ramakanth Illa, Shankara Gayathri Radhakrishnan, B. Ratna Sunil, Ladislav Ducsay

Dear Authors,

I found many details that need to be improved or corrected. I described all these details in the reviewer's comments in the pdf file.

After taking them into account, I recommend resubmitting the manuscript for a final review.

Required corrections:

Title

Please let me know why the authors use the term nano/fertilizers with a slash. In the studied literature, the term nano-fertilizers or simply nanofertilizers is often used.

Please consider changing the title to:
Significance of Phosphate Nano-Fertilizers Foliar Application: Real-Field Study of Quantitative and Physiological Parameters of Sunflower Crop

Abstract

Row 31 and many others: Please change nano/fertilizers to nano-fertilizers throughout the manuscript.

Row 35 and many others: Please check the name hydroxylapatite. It should be hydroxyapatite. Is it correct? Please confirm.

Row 35: Please change the abbrev. nano-HAP to nano-Hap throughout the manuscript.

Keywords

Row 48: Please change nano/fertilizers to nano-fertilizers, Please check the name hydroxylapatite. It should be hydroxyapatite.

1.    Introduction

Row 54: Before using the abbreviation please explain means: nano fertilizers (NFs)

Row 116: Before using the abbreviation please explain means: hydroxyapatite (HAP)

Row 142: Please use the defined abbreviations throughout the manuscript NFs

2.    Material and Methods

Row 147: Please add subsection: 2.1.1. Formation of hydroxyapatite

Row 155: Please add subsection: 2.1.2. Formation of mixture nanosized calcium zinc phosphate and macro-sized parascholzite

Row 170, 172: Please add equipment producer as follow (Name, City, State, Country) (Bruker, Billerica, MA, USA).

Row 270: Table 1a, 1b please use mg kg-1 without slash

Row 220: MDPI journals strongly recommend the use of SI units, with the exception of 1 ha which is accepted as a worldwide unit of area. Please change L to dm3

Row 223: There is no clear definition of treatments in the methodology. Authors must write how the individual objects of the experiment were treated.
15-15-15 DUSLOFERT fertilizer was applied to all objects, but in what dose?
3 series of experiment were used:
1) without nano-fertilizers,
2) nano-hydroxyapatite (nano-HAp)
3) nano-calcium zinc phosphate + macro-sized parascholzite (nano/macro-ZnPhos). Please explain it clearly in the methodology.

Row 230: Please specify for what purpose the ultrasonication was carried out?

Row 231: Days after sowing or days after emergence? Did the authors plant sunflowers?

Row 240: Please check the syntax error of this sentence.
Please check again. See line 362 and comments to table 4a and 4b. Should it be 100,000 μm2?

Row 245: Please add (Producer/manufacturer, City, Country)

Row 252: Please use abbrev. CA

Row 256: (CLAAS GmbH & Co.,Harsewinkel, Germany) please confirm.

Row 264: megagrams per hectare (Mg ha-1).

Row 270: Please add an explanation of what means R740 and R660 as follows: where R740 is absorption near infra-red wavelength and R660 is the absorption of red wavelength regions

Row 281: Eq 2 and 3, Please change superscripts to subscripts

Row 287: dm3

3.    Results

Row 312: Figure 4b. Please add a description of the Y-axis

Row 321: Table 2, Please check the formula: Ca10(PO4)6(OH)2 Please confirm. (source: https://www.nature.com/articles/s41598-021-91064-y)

Row 328: Figure 5b. Please add a description of the Y-axis

Row 361: Please change Table 4a title. Shouldn't the entries marked with red color be in the footer below the table?

Row 362, 367 Why did the authors give the number of stomata per 100,000 μm2 and per 1 cm2? Isn't it enough to specify per 100,000 μm2?
Table 4a and 4b I suggest removing this table row Number of stomata per 1 mm2

Row 386: Please center text in rows: Quantitative parameters and Nutritional parameter

Row 397: Days after sowing or days after emergence? Did the authors plant sunflowers?

Row 410, 667: Please consider deleting this part of the manuscript. The results described earlier are sufficient for this publication. I propose to use the results on agrobiodiversity in another publication on the impact of nanofertlizers on the species composition of various epigeic terrestrial fauna.

5. Conclusions

Row 730: Please see the comments (rows 410, 667).

The review is done partially. After making needed corrections according to the reviewer’s suggestions please resend the manuscript for final review.

Yours sincerely

Reviewer

Author Response

(The authors gave the same response as above.)

Reviewer 5 Report

Dear Authors,

I trust this message finds you in good health and high spirits. I am writing to convey my appreciation for your manuscript titled “Significance of Phosphate Nano/fertilizers Foliar Application: Real-Field Study of Quantitative, Physiological Parameters, and Agro-Ecological Diversity in Sunflower.” Having thoroughly reviewed your work, I must commend your team for producing a highly detailed and comprehensive study.

The subject matter of foliar fertilization with nanoparticles is of utmost relevance in the current agricultural landscape, and your research adds significant value to this field. Your dedication and hard work are truly commendable, and I extend my heartfelt congratulations to your entire working group.

One aspect that stood out to me was the exceptional quality of your introduction. It serves as an excellent foundation for the manuscript, providing a clear and concise context for your study. Furthermore, your meticulous description of the materials and methods employed in your research is truly commendable. It leaves no room for ambiguity and greatly enhances the reproducibility of your work.

I would like to offer a suggestion for improvement regarding Table 1a and 1b. I believe that merging these two tables into a single Table 1, encompassing both years and altering the table title to something like "Soil Agrochemical Results for Both Experimental Years: Autumn 2021 and Spring 2022," would not only streamline the presentation but also enhance the visual appeal of your manuscript.

Your results section is exceptionally well-written and presents intriguing findings. I commend you for the insightful discussion section, which adeptly ties your results to fundamental processes and aligns them with existing research findings. This synthesis greatly enriches the overall quality of your manuscript.

Finally, your conclusion section is a testament to your research prowess. It succinctly summarizes the key takeaways from your study and leaves readers with a clear understanding of the significance of your work.

In conclusion, I would like to extend my heartfelt congratulations to you and your team for the outstanding work you have done. It was a pleasure to read your manuscript, and I am confident that your research will make a valuable contribution to the field of agricultural science.

Warm regards,

Author Response

Thank you for your supportive position during the manuscript submission.

Round 2

Reviewer 1 Report

The manuscript is good for publication.

Minor editing of English language required

Author Response

We agree, corrections to the English were performed in various places in introduction section according to reviewers’ suggestion.

Reviewer 2 Report

This manuscript should be no accepted because the conclusions are based on one field experiment (one site/one environment). The hypothesis must be tested in at least two years of field experiments owing to the potential effect of the interactions owing to environment, management, and genetic factors. These kinds of new technologies must be rigorously tested before it's recommended for use in field conditions.

Author Response

We possess a comprehensive grasp of the established norms governing conventional agronomic research. Nevertheless, owing to the substantial value-added and innovative nature of our research, characterized as a “type of short communication” we have decided to publish one-year results to quick disseminate information in the academic community. Additionally, we can offer have several of arguments why we select one-year trial: i) there currently exists no rigorous European or regional legislation governing one-year proof-of-concept and dissemination in-field agronomy experiments; ii) numerous manuscripts have recently emerged in the field of Agronomy, presenting one-year outcomes, as evidenced by the works of Jia et al. (2023), Asirifi et al. (2021), Choo et al. (2020), and, furthermore, in the Agriculture journal, there have been notable publications by Lima et al. (2021) and Schmidt et al. (2015). Additionally, it is noteworthy that our research group has made significant contributions to the academic literature, with several highly-cited manuscripts featured in MDPI journals, including those authored by Kolenčík et al. (Kolenčík et al. 2019, Kolenčík et al. 2022, Kolenčík et al. 2021, Kolenčík et al. 2020), as well as the recent work of Ernst et al. (2023); iii) our extensive agronomic impact, characterized by favorable trends in nano-fertilizer performance within the context of field experiments, has been firmly established since 2018. These findings encompass a broad spectrum of agronomic parameters and a multitude of outcomes. Our endeavor to conduct comprehensive research is also evident in the present manuscript, which has been substantiated by four positive reviewer responses; iv) additionally, manuscript preparation reacts to main editor requirements directly pointed to special issue "New Perspectives on Phosphorus Management in the Soil-Plant System-Looking for Solutions to the P Scarcity".

References

Jia, Z., Wu, B., Wei, W., Chang, Y., Lei, R., Hu, W. and Jiang, J. (2023) Effect of Plastic Membrane and Geotextile Cloth Mulching on Soil Moisture and Spring Maize Growth in the Loess–Hilly Region of Yan’an, China.

Asirifi, I., Werner, S., Heinze, S., Saba, C.K.S., Lawson, I.Y.D. and Marschner, B. (2021) Short-Term Effect of Biochar on Microbial Biomass, Respiration and Enzymatic Activities in Wastewater Irrigated Soils in Urban Agroecosystems of the West African Savannah.

Choo, L.N., Ahmed, O.H., Talib, S.A., Ghani, M.Z. and Sekot, S. (2020) Clinoptilolite Zeolite on Tropical Peat Soils Nutrient, Growth, Fruit Quality, and Yield of Carica papaya L. cv. Sekaki.

Lima, J.R., Goes, M.D., Hammecker, C., Antonino, A.C., Medeiros, É.V., Sampaio, E.V., Leite, M.C., Silva, V.P., de Souza, E.S. and Souza, R. (2021) Effects of Poultry Manure and Biochar on Acrisol Soil Properties and Yield of Common Bean. A Short-Term Field Experiment.

Schmidt, H.P., Pandit, B.H., Martinsen, V., Cornelissen, G., Conte, P. and Kammann, C.I. (2015) Fourfold Increase in Pumpkin Yield in Response to Low-Dosage Root Zone Application of Urine-Enhanced Biochar to a Fertile Tropical Soil, pp. 723-741.

Kolenčík, M., Ernst, D., Komár, M., Urík, M., Šebesta, M., Dobročka, E., Černý, I., Illa, R., Kanike, R., Qian, Y., Feng, H., Orlová, D. and Kratošová, G. (2019) Effect of foliar spray application of zinc oxide nanoparticles on quantitative, nutritional, and physiological parameters of foxtail millet (Setaria italica L.) under field conditions. Nanomaterials 9(11), 1559.

Kolenčík, M., Ernst, D., Komár, M., Urík, M., Šebesta, M., Ďurišová, Ľ., Bujdoš, M., Černý, I., Chlpík, J., Juriga, M., Illa, R., Qian, Y., Feng, H., Kratošová, G., Barabaszová, K.Č., Ducsay, L. and Aydın, E. (2022) Effects of foliar application of ZnO nanoparticles on lentil production, stress level and nutritional seed quality under field conditions. Nanomaterials 12(3), 310.

Kolenčík, M., Ernst, D., Komár, M., Urík, M., Šebesta, M., Ďurišová, Ľ., Bujdoš, M., Černý, I., Juriga, M., Illa, R., Qian, Y., Feng, H., Kratošová, G., Šimonová, Z., Ducsay, L. and Aydin, E. (2021) Effect of foliar application of ZnO nanoparticles to lentil production, physiology, and nutrients seed quality at field conditions. Nanomaterials (v recenznom konaní).

Kolenčík, M., Ernst, D., Urík, M., Ďurišová, Ľ., Bujdoš, M., Šebesta, M., Dobročka, E., Kšiňan, S., Illa, R. and Qian, Y. (2020) Foliar application of low concentrations of titanium dioxide and zinc oxide nanoparticles to the common sunflower under field conditions. Nanomaterials 10(8), 1619.

Ernst, D., Kolenčík, M., Šebesta, M., Ďurišová, Ľ., Ďúranová, H., Kšiňan, S., Illa, R., Safarik, I., Černý, I., Kratošová, G., Žitniak Čurná, V., Ivanič Porhajašová, J., Babošová, M., Feng, H., Dobročka, E., Bujdoš, M., Pospiskova, K.Z., Afzal, S., Singh, N.K., Swamiappan, S. and Aydın, E. (2023) Agronomic investigation of spray dispersion of metal-based nanoparticles on sunflowers in real-world environments. Plants 12(9), 1789.

Reviewer 3 Report

My comments included in the review have been taken into account.

Comments from other reviewers allowed us to significantly improve the manuscript.

I recommend the article for inclusion in the Agronomy journal in its current form.

Author Response

Thank you for your support.

Reviewer 4 Report

Review Report 2

Manuscript ID: agronomy-2599185

Title: Significance of Phosphate Nano/fertilizers Foliar Application: Real-Field Study of Quantitative, Physiological Parameters, and Agro-Ecological Diversity in Sunflower

Authors: Dávid Ernst, Marek Kolenčík, Martin Šebesta, Ľuba Ďurišová, Samuel Kšiňan, Lenka Tomovičová, Nikola Kotlárová, Mária Kalúzová, Ivan Černý, Gabriela Kratošová, Veronika Žitniak Čurná, Jana Ivanič Porhajašová, Mária Babošová, Edmund Dobročka, Yu Qian, Sasikumar Swamiappan, Ramakanth Illa, Shankara Gayathri Radhakrishnan, B. Ratna Sunil, Ladislav Ducsay

Dear Authors

Most of my comments have been taken into account by you. I found a few more details that need to be improved. I described these details in the reviewer's comments, in a pdf file.

After taking them into account, I recommend submitting the manuscript for printing in the MDPI Journal - Agronomy.

Yours sincerely

Reviewer

Author Response

Thank you for the notice. We accept most of the comments from the attached pdf file. Changes can be checked in the latest version of the manuscript. We disagree with the addition of the manuscript’s title.